# Say My Name: a Model's Bias Discovery Framework

**Massimiliano Ciranni**[*]                                      *massimiliano.ciranni@edu.unige.it*
*MaLGa, DIBRIS, University of Genoa, Italy*

**Luca Molinaro**[*]                                              *l.molinaro@unito.it*
*Computer Science Department, University of Turin, Italy*

**Carlo Alberto Barbano**                                         *carlo.barbano@unito.it*
*Computer Science Department, University of Turin, Italy*

**Attilio Fiandrotti**                                            *attilio.fiandrotti@unito.it*
*Computer Science Department, University of Turin, Italy*
*LTCI, Télécom Paris, Institut Polytechnique de Paris, France*

**Vittorio Murino**                                               *vittorio.murino@iit.it*
*Istituto Italiano di Tecnologia (IIT), Genoa, Italy*
*University of Verona, Italy*

**Vito Paolo Pastore**[†]                                         *vito.paolo.pastore@unige.it*
*MaLGa, DIBRIS, University of Genoa, Italy*
*Istituto Italiano di Tecnologia (IIT), Genoa, Italy*

**Enzo Tartaglione**[†]                                           *enzo.tartaglione@telecom-paris.fr*
*LTCI, Télécom Paris, Institut Polytechnique de Paris, France*

**Reviewed on OpenReview:** *https://openreview.net/forum?id=EuUxiiDZ7d*

## Abstract

Due to the broad applicability of deep learning to downstream tasks and end-to-end training capabilities in the last few years, increasingly more concerns about potential biases to specific, non-representative patterns have been raised. Many works focusing on unsupervised debiasing leverage the tendency of deep models to learn "easier" samples, for example, by clustering the latent space to obtain bias pseudo-labels. However, their interpretation is not trivial as it does not provide semantic information about the bias features. To address this issue, we introduce "Say My Name" (SaMyNa), a tool to identify semantically potential biases within deep models. Unlike existing methods, our approach focuses on concepts learned by the model, enhancing explainability through a text-based pipeline. Applicable during either training or post-hoc validation, our method can disentangle task-related information and propose itself as a tool to discover biases. Evaluation on typical benchmarks demonstrates its effectiveness in detecting biases and even disclaiming them. When sided with a traditional debiasing approach for bias mitigation, it can achieve state-of-the-art performance while having the advantage of associating a semantic meaning with the discovered bias. The code is available at `https://github.com/SayMyName-BiasNaming/samyna-tmlr`.

---

[*]Equally contributing first authors
[†]Equally contributing senior authors

# 1 Introduction

In the past decade, advances in technology have made it possible to widely use deep learning (DL) techniques, greatly impacting the computer vision field. By allowing systems to be trained end-to-end, DL offers potentially fast model deployability to solve complex problems, leading to fast progress and changing how we perceive and analyze visual information. Today, deep learning is applied in various real-world scenarios, such as self-driving cars (Zhang et al., 2022a), medical imaging (Yan et al., 2024), and augmented reality (Liu et al., 2020).

These huge possibilities come with potential pitfalls. One challenge to face when deploying DL solutions lies in guaranteeing that the model does not over-rely on specific patterns that are non-representative of the real-world data distribution (Ming et al., 2022; Izmailov et al., 2022). This is key for safety, fairness, and ethics (Tartaglione et al., 2023). To provide a simple example, when deploying solutions in autonomous driving, simple tasks like pedestrian detection can be implicitly solved by associating specific background elements (such as sidewalks or pedestrian crossings) with the presence of pedestrians. This reliance on environmental cues can act as shortcuts for the net-

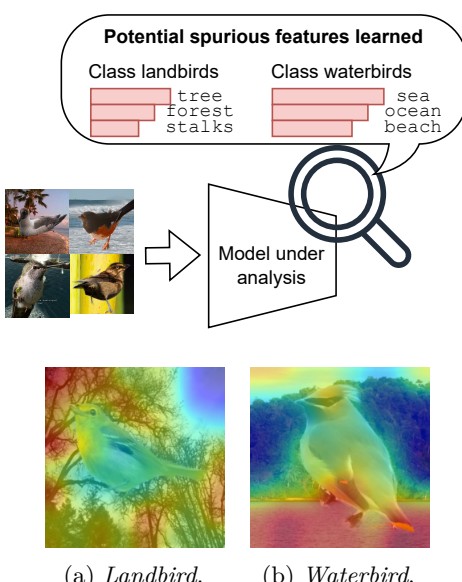

(a) *Landbird.* (b) *Waterbird.*

Figure 1: Top: SaMyNa searches potential spurious features learned by a model, providing a ranked list of keywords. Bottom: Example of bias heatmaps (red = correlated with bias, blue = anti-correlated).

work (Geirhos et al., 2020), leading to poor performance when the context changes, such as when a pedestrian is crossing a road without a marked crossing lane. DL models, if not discouraged, tend to rely on spurious correlations captured during training, especially when they are easier to learn than the actual semantic attributes (Nam et al., 2020). A visual example can be found at the bottom of Fig. 1, where a model trained on waterbirds, a dataset presenting a strong spurious correlation between target labels (waterbird and landbird) and background, indeed focuses on the latter for making its predictions. A description of how these heatmaps are constructed can be found in Sec. A of the Appendix. We refer to these spurious correlations as biases, and we say that the model that learned such shortcuts is *biased* towards them. In 2021, the European Commission introduced the Artificial Intelligence Act (AI Act) to regulate AI based on the potential risks it poses (Madiega, 2021). Like the General Data Protection Regulation (GDPR) (GDPR, 2016), the AI Act could set in the next few years a global standard. Ensuring that DL models avoid spurious biases that could affect their safety, trust, and accountability is not only essential for user safety but might soon also become a legal requirement.

A massive recent effort has been conducted by the Computer Vision community to try to discourage the presence of biases. In a nutshell, we can roughly distinguish three main research lines: (i) supervised, where the labels of the bias are provided (Barbano et al., 2023; Hong & Yang, 2021); (ii) bias-tailored, where a hint on what the potential bias might be is provided prior to training, and an ad-hoc model is deployed to capture it (Bahng et al., 2020b; Wang et al., 2019); (iii) unsupervised, where biases are guessed directly within the vanilla-trained model (Nam et al., 2020; Nahon et al., 2023; Creager et al., 2021; Li et al., 2022). When deploying a DL solution in the wild, the latter line of research appears to be the best fit, as detailed information about bias is almost surely missing. Although some solutions already exist in this context (Kim et al., 2024; Nam et al., 2020; Nahon et al., 2023; Ji et al., 2019), there is still a gap in the literature related to the problem of naming a specific bias affecting a DL model, providing natural language descriptors that can be directly interpretable by a human. Existing solutions either start from a predefined set of attributes (Eyuboglu et al., 2022; Wiles et al., 2023) or explicitly require the availability of a validation set known to contain bias-conflicting samples (Kim et al., 2024), and require to perform computationally expensive operations such as captioning the entire validation set, which may be unavailable or made up of only aligned samples.

Our method is effective by using only the training set, relaxing the assumptions of existing works (Kim et al., 2024), thus setting a first step towards mining specific model biases in realistic scenarios. Unlike approaches that discover biases in the dataset, our focus is on naming the biases captured by the model under examination. We mine the specific features in common with these samples and we associate semantic (textual) meaning to them. From this, an expert user (or prospectively a certifier software) can search and discriminate whether the learned feature is a bias or rather a feature for the system (Fig. 1). Through "Say My Name" (SaMyNa), we aim at providing a tool that enhances explainability for the DL model's learned features, on top of which, if necessary, any state-of-the-art debiasing approach can be used to sanitize the model.

Our contributions are here summarized at a glance.

- We propose a human-readable text-based pipeline for discovering and naming biases in DL models, which:
  - does not require a validation set, contrarily to existing approaches (Kim et al., 2024), and remains human-readable and interpretable at every step (Sec. 3.2);
  - leverages the embedding space of a text encoder to create learned class representations that capture potential biases of the model (Sec. 3.2.4);
  - can be applied both at training and inference time, with the former relying on a simple yet effective strategy for discovering potential biases of the model directly on the training set (Sec. 3.1).

- We validate our approach on established benchmarks, finding biases well-known by the community (Sec. 4.2.1). Furthermore, we also test on ImageNet-A, distinguishing cases where a bias exists and where, on the contrary, generalization issues are not bias-related (Sec. 4.2.2). This demonstrates the broad applicability of SaMyNa even for more general DL model diagnosis.

- After bias discovery (Sec. 3.1) and naming (Sec. 3.2), our method, when coupled with a standard debiasing strategy, can attain debiasing results in line with the state-of-the-art (Sec. 4.3), with the big advantage of assigning semantic meaning to the discovered bias.

## 2  Related Works

The problem of bias and model debiasing has been widely explored in recent years, within three main frameworks, differing from how or if bias knowledge is explored for mitigating model dependency on bias: supervised, bias-tailored, and unsupervised.

**Supervised Debiasing.**  Supervised methods require explicit knowledge of the bias, generally in the form of labels, indicating whether a sample presents a certain bias or not. One of the most typical approaches consists of training an explicit bias classifier, trained on the same representation space as the target classifier, in an adversarial way, forcing the encoder to extract unbiased representations (Alvi et al., 2018; Xie et al., 2017). Alternatively, bias labels can be exploited to identify pre-defined groups, training a model to minimize the worst-case training loss, thus pushing the model towards learning biased samples (Sagawa* et al., 2020). Another possibility is represented by regularization terms, which aim at achieving invariance to bias features (Barbano et al., 2023; Tartaglione et al., 2021).

**Bias-Tailored Debiasing.**  Bias-tailored approaches usually rely on some kind of knowledge about the bias nature. For example, if the bias is textural, then custom architectures can be designed to be more sensitive to textural information. For example, (Bahng et al., 2020a) propose ReBias, where a custom texture bias-capturing model is designed using 1x1 convolutions. A similar approach is followed by (Hong & Yang, 2021), where a BagNet-18 (Brendel & Bethge, 2019) is used as a bias-capturing model.

**Unsupervised Debiasing.**  Differently from the previously described approaches, unsupervised debiasing methods do not assume any prior knowledge of the bias, facing a more realistic situation where bias is unknown.

(Nam et al., 2020) propose LfF, where a vanilla bias-capturing model is trained with a focus on easier samples (bias-aligned), using the Generalized Cross-Entropy (GCE) loss (Zhang & Sabuncu, 2018), while a debiased network is trained by giving more importance to the samples that the bias-capturing model struggles to discriminate. (Ji et al., 2019) propose an unsupervised clustering method that learns representations invariant to some unknown or "distractor" classes in the data, by employing over-clustering. A set of unsupervised methods relies on the assumption that bias-conflicting samples are likely to be misclassified by a biased model (Kim et al., 2022a). In (Liu et al., 2021), a model is trained for a few epochs and then used in inference on the training set, considering misclassified samples as bias-conflicting and vice versa. The debiasing is then performed by up-sampling the predicted bias-conflicting samples. In (Kim et al., 2022a), the training set is split into a fixed number of subsets, training a model on each of them. Then, the trained models are ensembled into a *bias-committee* and the entire training set is fed to the committee, proposing that debiasing can be performed using a weighted ERM, where the weights are proportional to the number of models in the ensemble misclassifying a certain sample. Other lines of work, such as tackling training data containing noisy labels (Pleiss et al., 2020), introduced techniques to identify problematic training samples through margin-like measures. Similarly to (Nahon et al., 2023), we are able to identify during the training of a vanilla model in which moment the bias is potentially best fitted by the model, with the advantage of working directly at the output of the same model instead of mining the information in its latent space. This comes both with computational advantages (given that the latent space is typically higher dimensional) and with better interpretability of the outcome, given that we work in the model's output space.

**Bias Naming.** Recently, methods exploiting natural language and vision-language models to identify and mitigate bias have been proposed (Eyuboglu et al., 2022; Kim et al., 2024). (Zhang et al., 2023) introduce a method capable of determining subsets of images with similar attributes systematically misclassified by a model (i.e., error slices) and a rectification method based on language. However, it starts from a pre-defined set of attributes, thus hindering the possibility of discovering completely unknown and multiple biases. (Eyuboglu et al., 2022) exploits a cross-modal embedding space to identify error slices, providing natural language predictions of the identified slices. (Wiles et al., 2023) propose a method for automatically determining a model's failures, exploiting large-scale vision-language models and captioners to provide interpretable descriptors of such failures in natural language. In (Kim et al., 2024), the authors propose to extract class-wise keywords representative of bias, later used for model debiasing, exploiting group-DRO (Sagawa* et al., 2020) on the identified groups. In their work, a CLIP score is defined using the similarity between extracted keywords and correctly and incorrectly misclassified samples (class-wise) and used to find the keywords associated with a bias. In (D'Incà et al., 2024), the authors use large-language models and text prompts for bias discovery in text-to-image generative models. An alternative framework is proposed in (Guimard et al., 2025), where bias proposals are obtained relying on an LLM fed with task descriptions, including the textual name of the classes. A text-to-image retrieval approach, based on captions generated from the identified proposals, is then exploited to obtain biased images from an external large-scale dataset, further employed to evaluate the target model. Differently from the previously cited works, we introduce an unsupervised method for diagnosing a model dependency on bias for image classification tasks, which can either be performed during or after training and only exploits task-related knowledge to provide a transparent analysis of the potential hidden biases captured by the model.

## 3 Method

In this section, we present our proposed method SaMyNa to identify potential spurious correlations learned by the model under analysis (Sec. 3.2). Our method can perform model diagnosis either at training or at test time: for the first case, we also present an approach to identify at which iteration to mine a potential bias for the target model (Sec. 3.1).

### 3.1 Mining Model's Biases

Consider a supervised classification setup (having $C$ target classes), where we learn from a dataset $\mathcal{D}_{\text{train}}$ containing $N$ input samples $(\boldsymbol{x}_1, \ldots, \boldsymbol{x}_N) \in \mathcal{X}$, each with a corresponding ground truth label $(y_1, \ldots, y_N) \in \mathcal{Y}$. A deep neural network $\mathcal{M}$, trained for $t$ iterations, produces an output distribution $\hat{\boldsymbol{y}}_{t,n} \in \mathbb{R}^C$ over the $C$

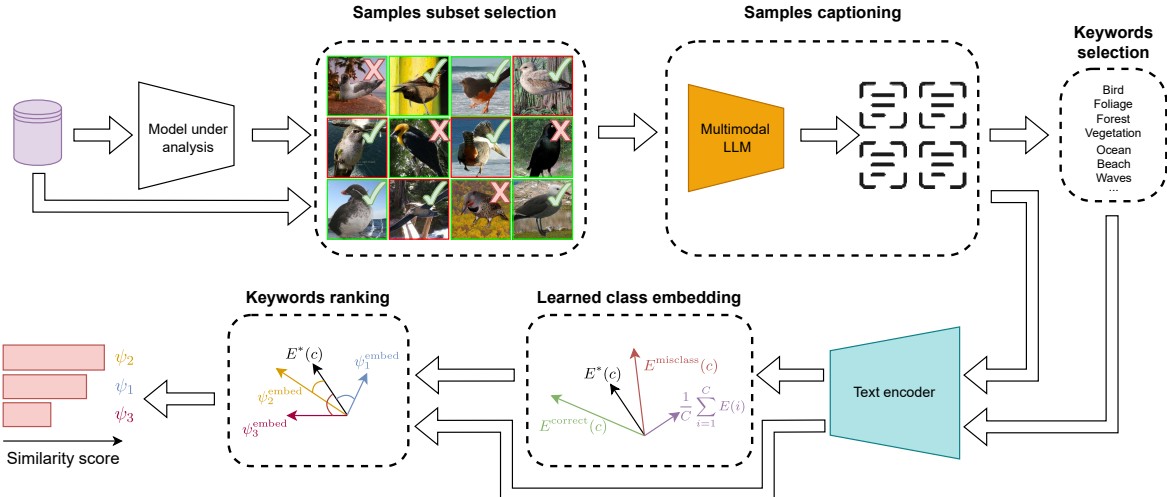

Figure 2: Pipeline for SaMyNa. Given a model, we can tell between either $\mathcal{D}^{\text{train}}$ or $\mathcal{D}^{\text{val}}$, which are the correctly (with green border) and the incorrectly (red border) classified samples. Amongst these, we first perform a sample subset selection looking at the latent space of the model under analysis and choosing through $k$-medoids the most representative samples for the learned class. Then, we employ a captioner to get a textual description of these samples. From these descriptions, we extract non-rare words as keywords, and, in parallel, working in the latent space of a text encoder, we extract the mean description for the learned classes, cleansed from common features within the dataset. We finally compare this representation with the embedding of the keywords, revealing learned correlations that extend beyond the target class signal.

classes, for each input $\boldsymbol{x}_n$ (typically, the activation of the last layer is a softmax). The network is trained to match the ground truth label $y_n$ by minimizing a loss function such as cross-entropy.

If any bias is present in the training set, however, the learning process could drive the model towards the selection of spurious features (Sagawa* et al., 2020; Nam et al., 2020; Bahng et al., 2020a), resulting in misclassification errors. Recent findings show that it is possible to identify the moment when the model best fits the bias in the training set (Nahon et al., 2023). We will formulate here the problem of identifying, at training time, when the trained model maximally fits a potential bias.

We say that $\mathcal{M}$ misclassifies the $n$-th sample at the $t$-th learning iteration if $y_n \neq \arg\max(\hat{\boldsymbol{y}}_{t,n})$. Focusing on this example, we know that by minimizing the loss function we aim at increasing the value of the $y_n$-th component of the model's output, denoted as $\hat{\boldsymbol{y}}_{t,n}(y_n)$, while decreasing all the others. To measure how far the model is from predicting the desired category, we can define a per-sample distance metric telling us how far the $n$-th sample is from being correctly classified:

$$d_{t,n} = \begin{cases} \max(\hat{\boldsymbol{y}}_{t,n}) - \hat{\boldsymbol{y}}_{t,n}(y_n) & \text{if } \arg\max_n(\hat{\boldsymbol{y}}_{t,n}) \neq y_n \\ 0 & \text{otherwise.} \end{cases} \quad (1)$$

To give an intuition, the distance-metric formalized in equation 1 is inspired by the Hinge Loss (Rosasco et al., 2004), and could actually be formulated directly as $\max(\hat{\boldsymbol{y}}_{t,n}) - \hat{\boldsymbol{y}}_{t,n}(\hat{y}_n)$, as we use one-hot-encoded vectors for labels. Here, for each sample, we define the target class as the category for which the model yields the highest prediction confidence, regardless of its correctness. We then compute a per-sample metric as the margin between the confidence in this target class and the confidence in the actual ground-truth class. This value quantifies the confidence gap the model must overcome to make a correct prediction. By definition, this metric is zero when a sample is correctly classified and positive otherwise.

Intuitively, the higher $d_{t,n}$ is, the most $\mathcal{M}_t$ is confident in misclassifying $\boldsymbol{x}_n$ as a specific class. We are interested in finding the iteration $t^*$ such that the model most confidently misclassifies a pool of samples, as

in equation 2,

$$t^* = \arg\max_t \frac{1}{|\mathcal{D}_{\text{train}}^{\text{misclass}}|} \sum_n d_{t,n},$$ (2)

where $\mathcal{D}_{\text{train}}^{\text{misclass}}$ is the set of misclassified training samples, and $|\mathcal{D}_{\text{train}}^{\text{misclass}}|$ is its cardinality. When reaching $t^*$, the most informative samples are the misclassified ones: given that the model is most confident in misclassifying them, the model may have learned some spurious features deeply affecting its final predictions. Unlike prior works (Nahon et al., 2023) that speculate that misclassified samples embody a bias (with the goal of applying debiasing methods), our goal is to understand why these samples are misclassified, ultimately providing the end user of the system the possibility to acknowledge the presence of a bias. For the model $\mathcal{M}_t$ we will split the training dataset in a pool of correctly classified samples $\mathcal{D}_{\text{train}}^{\text{correct}}$ and misclassified $\mathcal{D}_{\text{train}}^{\text{misclass}}$. Examples of the vanilla model's output softmax distribution during training can be found in the Appendix (Sec. C.1).

## 3.2 Bias Naming

We present here SaMyNa, our bias naming approach. Fig. 2 proposes an overview of the main pipeline we used, consisting of the following steps:

1. *Samples subset selection.* Given that the objective of our proposed method is to identify biases learned by $\mathcal{M}$, we are allowed to propose a subset of most representative samples for a given target class (Sec. 3.2.1).

2. *Samples captioning.* We then generate textual descriptions for the selected samples (Sec. 3.2.2).

3. *Keywords selection.* Starting from the captions, we construct a pool of keywords by applying a lightweight word frequency filter. Heavy filtering is done as a last step to better account for synonyms (Sec. 3.2.3).

4. *Learned class embedding.* Simultaneously, we generate embeddings for the captions. We use these embeddings, along with the corresponding target and predicted classes, to calculate learned class embeddings that represent the biases of $\mathcal{M}$ if present (Sec. 3.2.4).

5. *Keywords ranking.* We then embed the keywords and rank them according to their similarity with the learned class embeddings. Finally, low-ranking keywords can be filtered out (Sec. 3.2.5).

### 3.2.1 Samples Subset Selection

Given $\mathcal{M}$, for a given target class $c$, we extract the pool of correctly classified samples $\mathcal{D}^{\text{correct}}(c)$ and the pool of samples misclassified as class $c$, which we call $\mathcal{D}^{\text{misclass}}(c)$.[1] Provided that $\mathcal{M}$ clusters both $\mathcal{D}^{\text{correct}}(c)$ and $\mathcal{D}^{\text{misclass}}(c)$ together, our hypothesis is that these two share a common set of features, behind which we might find a bias. In the typical deployment scenario, the correctly classified examples are abundant, and, for instance, $\mathcal{M}$ may project them in a very narrow neighborhood of its latent space. We build on top of this observation, and we cluster correctly classified samples using the $k$-medoid algorithm to reduce the cardinality of correctly classified samples while maintaining a good coverage of the model's latent space. The choice of using $k$-medoid is natural, since it provides samples as output instead of centroids (which may be arbitrary points in feature space). As our long-range objective is to capture the set of features learned by the model to make its predictions, we also select a sample of examples misclassified as class $c$ by sorting them according to $d_{t^*,n}$, selecting the $k$ ones that were misclassified with more confidence. We refer to the $k$ samples resulting from running $k$-medoid on the correctly classified samples as $\mathcal{S}^{\text{correct}}(c)$, while the pool of $k$ samples misclassified as $c$ is denoted as $\mathcal{S}^{\text{misclass}}(c)$.

---

[1]please note that we have dropped the "train" subscript from the $\mathcal{D}$ partitioning as this pipeline will work equally well also when using a validation set.

### 3.2.2 Samples Captioning

At this point, we will generate captions from the selected samples. To do this, we use a pre-trained multimodal large language model that takes as input both a prompt and an image. We used the following generic captioning prompt for all the experiments: "`Describe this image in detail. Pay particular attention to the subject and where they are. Describe as many details as possible`".

We decided to employ a large-scale image captioner in all our experiments, given that biases might hide in the subtle characteristics of the provided images (see Sec. B of the Appendix).

### 3.2.3 Keywords Selection

From the captions obtained in the previous step (Sec. 3.2.2), we select candidate keywords. First, we perform some NLP standard processing, consisting of lower-case conversion, word-level tokenization (Bird et al., 2009), and stop-words removal. Then, for each token $w$ and class $c$, we count the number $f_w(c)$ of captions of class $c$ that contain $w$ at least once. Lastly, given $f_{\min}$ between 0 and 1, we filter out the token $w$ if $\max_c[f_w(c)]$ is less than $f_{\min} \cdot [|\mathcal{S}^{\text{correct}}(c)| + |\mathcal{S}^{\text{misclass}}(c)|]$. The remaining tokens are aggregated into a keywords proposal pool $\Psi$.

### 3.2.4 Learned Class Embedding

In parallel to keywords selection, we aim at having a representation for the learned class, disentangled from the specific domain $\mathcal{M}$, which is trained to. To do this, we work in the embedding space of a pre-trained text encoder. From the generated captions we obtain, for each class $c$, the embedding matrices $E^{\text{correct}}(c) \in \mathbb{R}^{|\mathcal{S}^{\text{correct}}(c)| \times Z}$ and $E^{\text{misclass}}(c) \in \mathbb{R}^{|\mathcal{S}^{\text{misclass}}(c)| \times Z}$, where $Z$ is the dimensionality of the embedding vectors. These matrices are simply the results of stacking the embedding vectors coming from the text encoder applied to the various captions of the samples in $\mathcal{S}^{\text{correct}}(c)$ and $\mathcal{S}^{\text{misclass}}(c)$. Our goal here is to calculate the embeddings $E^*(c) \in \mathbb{R}^Z$ semantically representing the learned representation of the class $c$. Not only that, we would like to disentangle this from the features in common to all the classes learned from the model, given that $\mathcal{M}$ could be trained (and tested) to fit a specific domain, which in such a specific case would not constitute a bias but rather a feature. For this, we first calculate the embedding $E(c)$ for a specific learned class:

$$E(c) = \frac{\sum_{i=1}^{|\mathcal{S}^{\text{correct}}(c)|} \sum_{j=1}^{|\mathcal{S}^{\text{misclass}}(c)|} [E_i^{\text{correct}}(c) + E_j^{\text{misclass}}(c)]}{2\left[|\mathcal{S}^{\text{correct}}(c)| \cdot |\mathcal{S}^{\text{misclass}}(c)|\right]}. \tag{3}$$

This equation computes the average of the means across all pairs of embeddings, where each pair consists of a correctly classified example of class $c$ and an example that was misclassified as class $c$. We adopt this pairwise formulation instead of a simple average to improve robustness in scenarios with significant imbalance between correctly and incorrectly classified samples. This imbalance usually happens when $k$ is greater than the number of misclassified examples for a given class. Additionally, because each pair includes embeddings from examples with different ground-truth classes, this approach helps discourage the encoding of the target class into the learned class embedding. $E(c)$ will now contain all the common features of the class $c$. However, it will also contain some information shared in the entire dataset (for example, common characteristics of the different classes). From this, we can extract the embedding without the shared information from the dataset through:

$$E^*(c) = E(c) - \frac{1}{C} \sum_{i=1}^{C} E(i). \tag{4}$$

The intuition of this approach originates from the arithmetic and semantic properties of natural language latent spaces (Mikolov et al., 2013). To provide a realistic example, consider the task of gender recognition from facial pictures, in which the hair color is a spurious correlation. $E(c)$ might contain features related to concepts such as "blonde" and "face". As we are only interested in the former, computing $E^*(c)$ is an effective solution to filter out the shared information "face". We demonstrate the effectiveness of equation 4 through an ablation study in Sec. C.2.1 of the Appendix.

### 3.2.5 Keywords Ranking

Now, we are ready to compare the embedding of each keyword with $E^*(c)$ using the cosine similarity:

$$s(\psi, c) = \text{sim}[\psi^{\text{embed}}, E^*(c)], \quad \psi \in \Psi, \tag{5}$$

where $\psi^{\text{embed}}$ is the embedding of the keyword in the same latent space used to calculate $E^*(c)$. This tells us how much the concept is embodied by the proposed keywords. Based on the ranking, we will obtain a set of keywords that correlate with the learned class $c$, and others that become decorrelated as they embody some knowledge shared through all the classes (as filtered in equation 4). For this, we introduce a hyper-parameter $t_{\text{sim}} > 0$ that thresholds the relevant keywords for the learned class $c$, based on the similarity score. The final ranking we obtain embodies the set of features that correlate with the learned class $c$, from which an end user of the system can deduce the presence of a bias. In some cases, the ranking may also contain keywords related to the target class, but such keywords are easily ignored by the end user.

## 4 Empirical Results

We provide here the main results obtained. We highlight that, for visualization purposes, all the figures contain up to the top nine keywords identified by SaMyNa: additional ablation studies and the full results are presented in Sec. C and H of the Appendix, respectively. For our experiments, we have employed an NVIDIA A5000 with 24GB of VRAM, except for the captioning step for which we have employed an NVIDIA A100 equipped with 64GB of VRAM. The source code, attached to the submission, will be open-sourced upon acceptance of the article.

### 4.1 Setup

**Models tested.** We tested the most popular architectures benchmarked from the debiasing literature: ResNet-18 for CelebA and BAR, and ResNet-50 for Waterbirds and ImageNet-A. All the models are pre-trained on ImageNet-1K, with architecture and weights provided by `torchvision`. On ImageNet-A, models are run only in inference, as we are interested in mining biases already existing in the original pre-trained models, while for all the other experiments we apply the training procedure described in Sec. 3.1. For this step, we train with a batch size of 128 and a learning rate of 0.001 for Waterbirds, as done in (Sagawa* et al., 2020); for CelebA, we use a batch size of 256 and a learning rate of 0.0001, following (Nam et al., 2020). For both, we employ SGD with Nesterov, set to 0.9. Finally, for BAR, we employ a batch size of 256 and a learning rate of 0.001, with Adam as the optimizer (Kim et al., 2021).

**Captioning.** For the captioning model, we used LLaVA-NeXT (Liu et al., 2024)[2] in its 34B configuration, quantized in 8 bits. Before feeding our input images to the image captioner, we apply the corresponding preprocessing transform provided by the huggingface library.

**Class and keywords embedding.** For this part, we used a MiniLM model[3] from the sentence-transformers (Reimers & Gurevych, 2019) library to generate 384-dimensional embeddings of the captions. The minimum frequency for the keywords $f_{\text{min}}$ is 15%. For the correctly classified samples, we set $k = 10$ and $t_{\text{sim}}$ is 0.2.

**Datasets.** For our study, we employ the following datasets: Waterbirds (Sagawa* et al., 2020), CelebA (Liu et al., 2015), BAR (Nam et al., 2020), and ImageNet-A (Hendrycks et al., 2021).

*Waterbirds* is an image dataset introduced in Sagawa* et al. (2020) to test the robustness of optimization methods against distribution shifts. The associated task is to classify the habitat of bird species, divided into *waterbirds* and *landbirds*. In the training set, though, 95% of waterbirds are associated with a water background, and 95% of landbirds are set on land. Only 5% of the two classes' samples are presented with an *opposite* background. This results in a potentially strong spurious relation between the target label and the background.

---

[2] https://huggingface.co/llava-hf/llava-v1.6-34b-hf
[3] https://huggingface.co/sentence-transformers/paraphrase-MiniLM-L12-v2

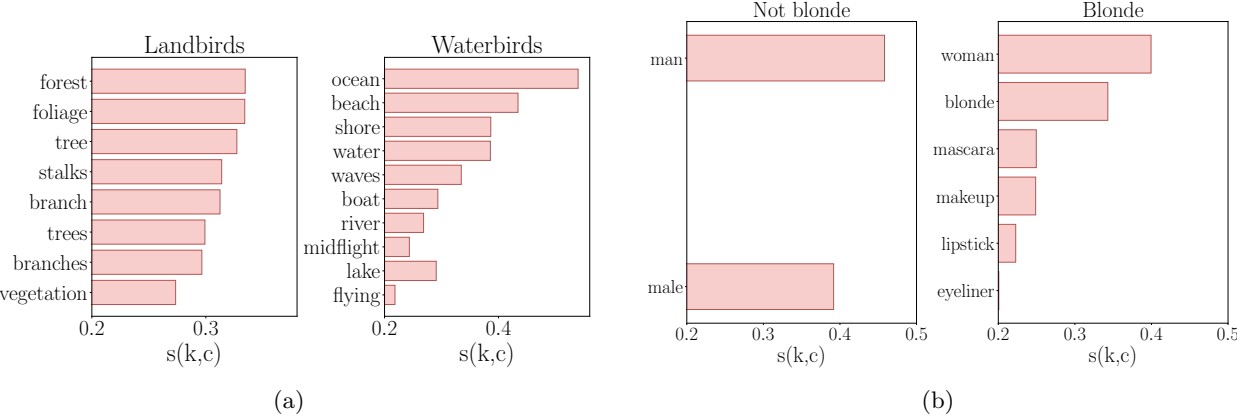

Figure 3: Similarity scores for Waterbirds (a) and CelebA (b).

*CelebA* (Liu et al., 2015) is one of the most popular datasets of face images, depicting celebrity individuals with multiple samples per subject and equipped with extensive annotations regarding roughly 40 attributes, encompassing several face and style characteristics. Regardless of its popularity, it is notoriously affected by several biases (Nam et al., 2020; Barbano et al., 2023). In this work, we analyze the task of classifying whether a person has blond hair or not, for which the attribute *female* is not uniformly represented, but spuriously correlated with the *blond* class. A deep network trained with standard techniques exhibits a strong bias in predicting subjects with female perceived gender as blond, while male appearing subjects are often classified as *not blond.*

*Biased Action Recognition* (*BAR*) is an action recognition dataset crafted by Nam *et al.* in Nam et al. (2020). The target classes of BAR (*Climbing, Diving, Fishing, Racing, Throwing, Vaulting*) present a strong bias towards the setting where they are performed. For instance, the large majority of samples from the class *Racing* is depicted in a circuit track context, while in the test set, we can find many *off-road* settings on which deep classification models fail to generalize. This dataset does not provide bias group annotations and does not provide a proper validation set, making it impossible to compute a Worst Group accuracy.

*ImageNet-A* is a collection of 7,500 real-world images sharing the same category of a 200-class subset of ImageNet, onto which deep models systematically fail to output the correct prediction. Originally introduced in Hendrycks et al. (2021) for testing adversarial model robustness, it is also commonly adopted in the context of model bias (Bahng et al., 2020b; Kim et al., 2022b).

### 4.2 Bias Discovery

Our empirical validation on bias discovery is here divided into naming the bias during training (Sec. 4.2.1) and naming it post-hoc, at inference (Sec. 4.2.2). In Sec. 4.3, we provide a description of how our keywords can be used to perform bias mitigation, and in Sec. 5.2 we show how SaMyNa behaves in an unbiased case.

#### 4.2.1 Naming the Bias at Training Time

We begin by discussing the results of SaMyNa when applied in a model's training on Waterbirds, CelebA, and BAR.

**Waterbirds.** The barplot in Figure 3a shows the candidate bias keywords for Waterbirds, alongside their relative similarity value. We can observe how the obtained keywords are mainly related to the background information (`forest` and `foliage` for landbirds; `ocean` and `beach` for waterbirds). Most importantly, the top keywords display high similarity values, indicating a high correlation with their class targets. We deduce that the model suffers from a bias about image backgrounds, which indeed is the case for Waterbirds. It is also worth noticing how the top similarities differ among the two classes. We hypothesize two possible factors causing it: (i) model bias towards `ocean` is stronger than the one towards `forest`, as it is constituted by

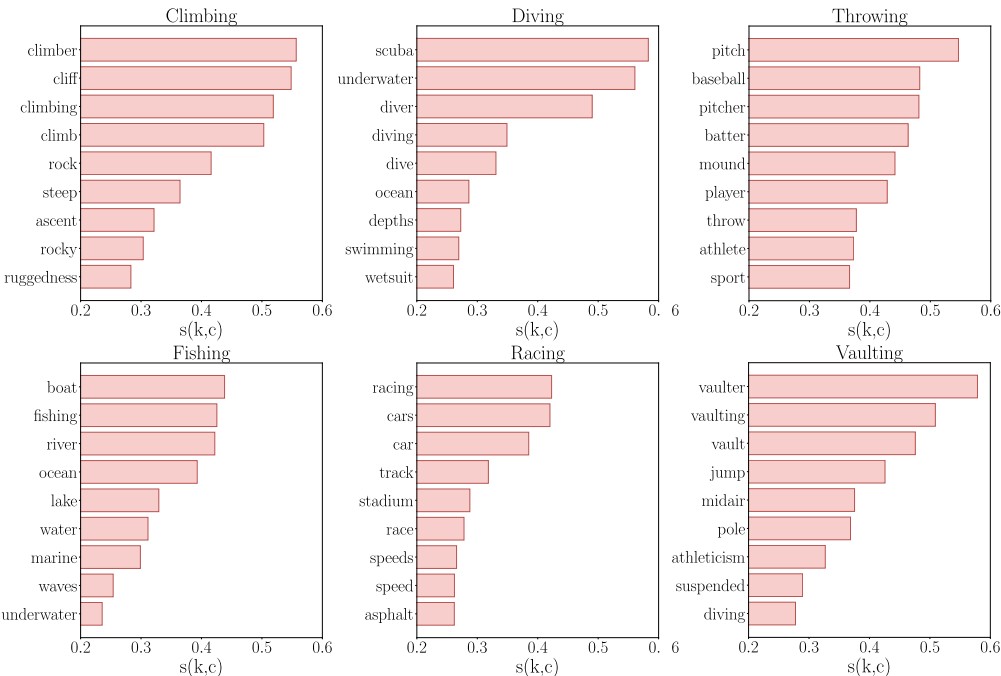

Figure 4: Similarity scores for the BAR dataset.

simpler visual patterns, easier to learn for the network; (ii) the *landbirds* class has a much larger population, thus allowing for the bias on this class to be averaged over more instances than the *waterbirds* case.

**CelebA.** In analyzing the possible biases of our vanilla model on CelebA (see Figure 3b), we find that the top-1 keyword for both classes represents a gender: `male/man` for class *not blonde* (with similarity $\approx 0.4$), and `woman` for class *blonde* (similarity of $\approx 0.4$). Additionally, we find among the *blonde* class, keyword terms typically associated with gender stereotypes, such as `mascara`, `makeup`, and `lipstick`, with moderately high similarity. This suggests that the vanilla model is not only biased regarding the classification task but also incorporates *unfair features*, arising from societal biases reflected in the training data.

**BAR.** Bar presents a more challenging situation, due to the complete absence of bias group annotations. Regardless, the output of our approach still provides insights about the biases captured by the model. In the training class *climbing*, scenes where the subject is ascending on rocks are overly represented, and thus the vanilla model has wrongfully learned to rely on the presence of rocky backgrounds. This is reflected by the top three keywords not related to the target class in Figure 4 (`cliff`, `rock`, and `steep`), all having high similarities. Similar considerations can be made on the other classes: `pitch`, `baseball`, `pitcher`, and `batter` suggest an even stronger bias for the class *throwing* towards a specific sport, baseball, which is also the second most correlated keyword. The same can be said for `scuba` and `underwater` in the *diving* class. Keywords for other classes show slightly lower maxima. Still, we can measure relevant correlations with the presence of cars and circuit races for *racing*, as well as boats and water (e.g. `rivers`, `ocean`, `lake`) for *fishing*. Complete outputs of the keyword extraction process are available in the Appendix.

### 4.2.2 Naming the Bias at Inference Time

To evaluate the capabilities of our method in describing potential biases at inference time, we design a dedicated experiment involving ImageNet-A. In particular, we are interested in finding specific model failures and extracting an interpretable set of keywords that can guide an expert practitioner to tackle them. With this aim, we first build an evaluation set as the union of the whole ImageNet-A dataset with the samples in the validation set of ImageNet-1K sharing the same 200 categories of ImageNet-A. Then, we run the model in inference over this dataset, collecting $\mathcal{D}^{\text{correct}}$ and $\mathcal{D}^{\text{misclass}}$ directly without any additional training. In this experiment, we are not interested in finding dataset-wise biases, but rather in assessing the behavior of

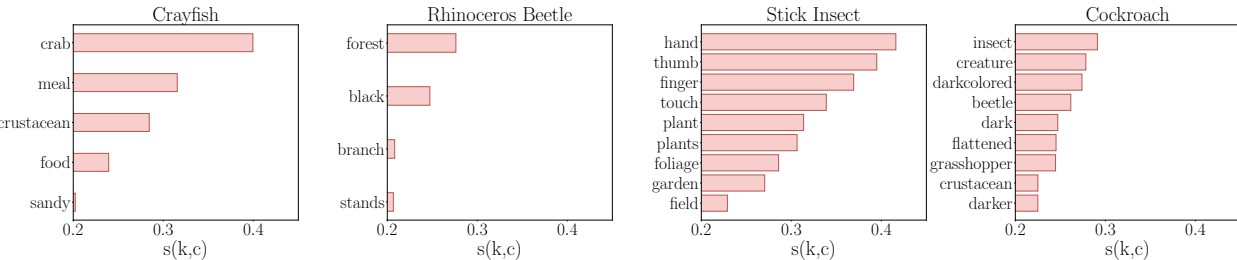

Figure 5: Similarity scores for the *crayfish*, *rhinoceros beetle*, *stick insect* and *cockroach* classes from ImageNet-A.

| Method | No Val.Set in Bias-Id | Unsup. | Bias Named | CelebA (Hair Color) | | Waterbirds | | BAR |
|---|---|---|---|---|---|---|---|---|
| | | | | Average | Worst Group | Average | Worst Group | Average |
| LISA Yao et al. (2022) | – | ✗ | ✗ | 92.40 | 89.30 | 91.80 | 89.20 | – |
| GroupDRO Sagawa* et al. (2020) | – | ✗ | ✗ | $92.90 \pm 0.20$ | $88.90 \pm 2.30$ | $93.50 \pm 0.30$ | $91.40 \pm 1.10$ | – |
| George Sohoni et al. (2020) | ✗ | ✓ | ✗ | 94.60 | $54.90 \pm 1.90$ | 95.70 | $76.20 \pm 2.00$ | – |
| JTT Liu et al. (2021) | ✗ | ✓ | ✗ | 88.10 | $81.50 \pm 1.70$ | 89.30 | $83.80 \pm 1.20$ | $68.53 \pm 3.29$ |
| CNC Zhang et al. (2022b) | ✗ | ✓ | ✗ | 89.90 | $88.80 \pm 0.90$ | 90.90 | $88.50 \pm 0.30$ | – |
| B2T+GDRO Kim et al. (2024) | ✗ | ✓ | ✓ | 93.20 | $90.40 \pm 0.90$ | 91.50 | $\mathbf{90.70 \pm 0.30}$ | $68.54 \pm 0.86^*$ |
| ERM | ✓ | ✓ | ✗ | **94.90** | $47.70 \pm 2.10$ | **97.30** | $62.60 \pm 0.30$ | $51.85 \pm 5.92$ |
| LfF Nam et al. (2020) | ✓ | ✓ | ✗ | 84.24 | $81.24 \pm 1.38$ | 91.20 | 78.00 | $62.98 \pm 2.76$ |
| DebiAN Li et al. (2022) | ✓ | ✓ | ✗ | 84.00 | $52.90 \pm 4.70$ | – | – | $69.88 \pm 2.92$ |
| SaMyNa (ours) + GDRO | ✓ | ✓ | ✓ | $92.20 \pm 0.01$ | $\mathbf{90.60 \pm 0.08}$ | $91.11 \pm 0.44$ | $\mathbf{90.62 \pm 0.10}$ | $\mathbf{71.26 \pm 1.46}$ |

Table 1: Performance of GDRO Debiasing on top of our training-time bias naming pipeline. Column *Unsup.* indicates if the method uses ground truth bias information. Best results for unsupervised debiasing methods are highlighted in bold. *No Val.Set in Bias-Id* highlights if the method does not rely on a validation set for inferring subgroups or bias-attributes (green tick, ✓) or it assumes having one (red cross, ✗). *Bias Named* indicates if the method extracts semantic names of found bias attributes. BAR lacks worst-group accuracy due to the absence of bias annotations. Average refers to the unbalanced test accuracy. * These results on B2T+GDRO were computed by us.

our approach in specific and challenging real-world scenarios. Hence, we derive a specific case study from the systematic model confusion obtained from its predictions, which could hide the presence of a possible model bias. Another analysis on ImageNet-A describing more general DL diagnosing features is provided in Sec. 5.3, while here we limit the discussion to the model bias perspective.

Our key study (see Figure 5) involves a subset of four classes: *crayfish*, *rhinoceros beetle*, *stick insect*, and *cockroach*. Samples (correctly or incorrectly) classified as *crayfish* are often placed in a setting depicting plates, tables, or people eating, thus the bias reflected by the keywords `meal` and `food`. Moreover, `crab` probably suggests the inability of the model to distinguish between the two closely related species. At the same time, for the classes *stick insect* and *rhinoceros beetle*, several samples show the creature being held in a person's hand, often in a vegetation setting (hence the keywords `hand`, `thumb`, ..., `plants`, `foliage`). From these observations, we validate the presence of a possible bias among these classes, for which the source of error is not caused only by the fine-grained nature of these categories. An additional analysis involving Vision Transformers is provided in Sec. 5.1.

## 4.3 Mitigating the Discovered Bias

Starting from the keywords that were extracted to describe potential bias affecting the classification model, we here validate if the attributes suggested by SaMyNa can be leveraged to improve our network's generalization capabilities.

**Setup.** After training-time bias identification and naming (SaMyNa), we exploit a pre-trained CLIP model as a zero-shot classifier (Radford et al., 2021) to infer the alignment of each sample towards the

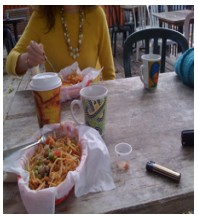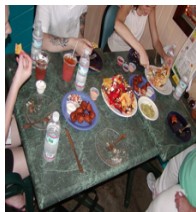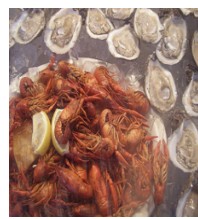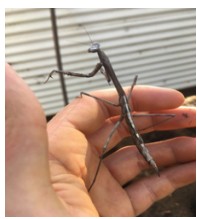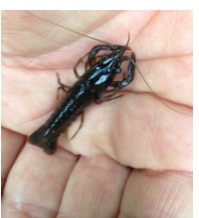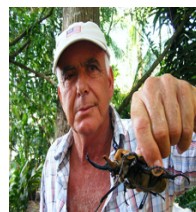

Figure 6: Misclassified samples from *crayfish* (three left-most pictures), and from the *insects* classes (three right-most pictures). It can be seen how the contexts in which the objects of interest appear reflect the found bias-keywords.

bias-keyword(s) of its target class or not. As a result, we obtain subgroup pseudo-labels, which can then be leveraged in a state-of-the-art supervised debiasing algorithm (e.g. GroupDRO (Sagawa* et al., 2020)).

For each dataset, we employ the following approach: given a training sample $(x_i, y_i)$, and the set of keywords found for its class $y_i$, we use CLIP's image and text encoders to obtain its embeddings $z_\text{img}$ and $z_\text{txt}$. A bias label is then computed for each sample by just annotating if the zero-shot classification from CLIP corresponds to a *bias-keyword* from its class or not. Finally, we plug our set of pseudo-labels in GroupDRO and measure the obtained test performances. In this stage, we employ the same hyperparameters used in the GroupDRO original implementation for CelebA and Waterbirds. For BAR, we set the learning rate and weight decay to $5 \times 10^{-4}$ and $10^{-3}$, respectively, with a batch size of 128. Group adjustment is set to zero, and $\alpha$ is set to 0.5.

**Results.** Table 1 outlines the obtained results. Here, we categorized existing work according to two main factors: (i) Unsupervised (Unsup.), i.e. if the method does not use ground truth bias information (✓) or it does (✗); (ii) No Val.Set in Bias-Id, to better highlight whether bias information inference relies on the usage of a validation set with bias annotations. Our semantic bias discovery-based approach outperforms all the unsupervised methods. Compared to the only other method able to name biases (B2T+GDRO), our approach achieves slightly higher worst-group accuracy on average. Since our method uses GroupDRO with automatically inferred pseudo-labels, its upper bound is supervised GroupDRO with ground-truth bias labels. On CelebA, our method exceeds this upper bound due to noisy group annotations in the dataset; on Waterbirds, where labels are synthetic and accurate, we closely match it.

While B2T+GDRO performs similarly to SaMyNa on CelebA and Waterbirds, its keywords describe the inverse of the bias (which is less interpretable), and on CelebA it detects only one of two spurious features—yet, for the binary case, this is sufficient for GroupDRO to work. Additionally, our method does not rely on any validation set (e.g. BAR does not have one), and the amount of image captions we require is quite limited, thanks to the exemplar-mining step described in Section 3.2.1, whereas B2T directly captions the entire validation set (for an in-depth comparison with B2T refer to Sec. D in the Appendix). This feature allows our Bias-Discovery method to be employed when bias annotations are not present at all, as in BAR. Indeed, if we extract a held-out validation set from BAR and run B2T+GDRO on it (see Tab. 20 in the Appendix for the extracted keywords), we measure an average drop in Average Accuracy of 3% compared to SaMyNa. Notably, since both our method and B2T rely on GroupDRO for the final bias mitigation step, our advantage has to be imputed on a finer semantic bias discovery.

# 5 Additional Analyses and Ablation Studies

## 5.1 Bias Discovery on Vision Transformer Models

We present here a study on two popular pre-trained Vision Transformers architectures: ViTb-16 and Swin-V2. Tab. 2 reports the outcome of SaMyNa for the classes *crayfish*, *rhinoceros beetle*, *stick insect*, and *cockroach* of ImageNet-A. Despite the potential of generalization for these architectures, we are still able to observe, although with different magnitudes, some biases. Regarding ViTb-16, the class *crayfish* is still associated with `meal`, and *cockroach* is associated with `floor`: interestingly, the impact of `hand` for the *stick insect* is heavily

| Tested architecture | Target | Keyword (value) |
|---|---|---|
| ViTb-16 | *Crayfish* | crayfish (0.50581), crab (0.48959), crustacean (0.37085), meal (0.35827), food (0.27727), plate (0.25765), water (0.22597) |
| | *Rhinoceros Beetle* | beetle (0.40866), tree (0.26703), forest (0.26600), trees (0.25613), branch (0.22754), spider (0.21616), insect (0.21564) |
| | *Stick Insect* | foliage (0.40032), plants (0.36978), plant (0.36220), garden (0.33877), grass (0.29491), leaves (0.28791), trees (0.28398), forest (0.27847), flowers (0.26615), tree (0.24779), field (0.22410), thumb (0.21469), hand (0.21322), green (0.21072), brown (0.20495) |
| | *Cockroach* | cockroach (0.34074), insects (0.27413), floor (0.26161), insect (0.25462), grasshopper (0.23879), glossy (0.23501), beetle (0.23324), darkcolored (0.23225) |
| Swin-V2 B | *Crayfish* | lobster (0.60554), crab (0.49117), crayfish (0.44054), crustacean (0.40692), underwater (0.34094) |
| | *Rhinoceros Beetle* | beetle (0.49412), butterfly (0.41977), insect (0.28676), insects (0.27957), wings (0.23552), forest (0.23534), trees (0.22520), nature (0.22299), grasshopper (0.21111), black (0.20005) |
| | *Stick Insect* | foliage (0.37396), plant (0.35905), garden (0.32784), leaves (0.28342), grass (0.27446), hand (0.27024), green (0.26462), trees (0.23665), touch (0.23344), lush (0.21824), forest (0.21634), stem (0.20281) |
| | *Cockroach* | cockroach (0.32120), floor (0.23421), uneven (0.20648) |

Table 2: Testing pre-trained Vision Transformer architectures on *crayfish*, *rhinoceros beetle*, *stick insect*, and *cockroach* classes from ImageNet-A.

reduced compared to the ResNet model, while with the introduction of sliding windows, it goes back up for Swin-V2. In general, we notice that these architectures, although still suffering from bias, are less prone to it, probably due to finer training enhanced by larger parametrization combined with the self-attention mechanism they embody.

## 5.2 SaMyNa on an Unbiased Dataset

We propose here a study on a virtually balanced version of the Waterbirds dataset. Fig. 7 reports the results in a graphical form, while Tab. 27 of the Appendix reports the numerical values. While we should have a priori removed the bias by balancing the dataset, resulting in a general, massive reduction of the similarity scores, we still observe some mild correlations arising, especially for the *waterbirds* class. Specifically, besides the `seagull` keyword evidencing a (potential) higher presence of seagulls in the data split, we still see some concepts like `sea` and `midflight` correlating with the learned class. This is expected: given that these features are easy to learn, the model still captures them, but the low similarity score indicates that it does not heavily rely on them. This shows that, despite balancing the dataset, some

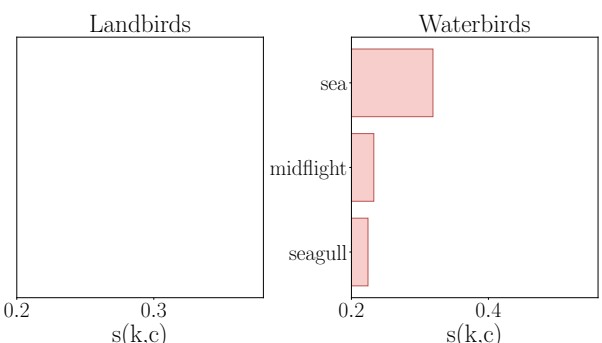

Figure 7: Ablation study on Waterbirds, where we balance the two classes, a-priori removing the bias.

biased features can still permeate through the model, depending on how easy they are to capture. This further motivates our work, focusing on model debiasing rather than dataset debiasing. At the same time, the absence of highly correlated keywords for the *landbird* class (which is naturally more variable) is somewhat expected due to the removal of the original spurious correlation with the target.

### 5.3 Another Case Study for ImageNet-A: *nails* vs *mushrooms*

Besides the study provided in Sec. 4.2.2 of the main paper, we present here another case study on the ImageNet-A dataset, comparing two other critical classes: *nails* and *mushrooms*. Indeed, also for this case, the tested ResNet-50 model presents a big error between these two classes: is there a bias involved? Running our SaMyNa (Fig. 8), we observe that indeed there is a big correlation towards certain concepts for the *nails* class; however, these hardly resemble biases, but rather features of the target class. Indeed, we can easily imagine that concepts like `metal`, `frame`, or `rusted` can be easily associated with the target *nails* class. At this point, where is this big confusion arising from? The answer comes from a visual inspection of the samples, wherein multiple cases the shape factor of the two classes is extremely similar (both show a bulge on top and a thinner body underneath), making the classification

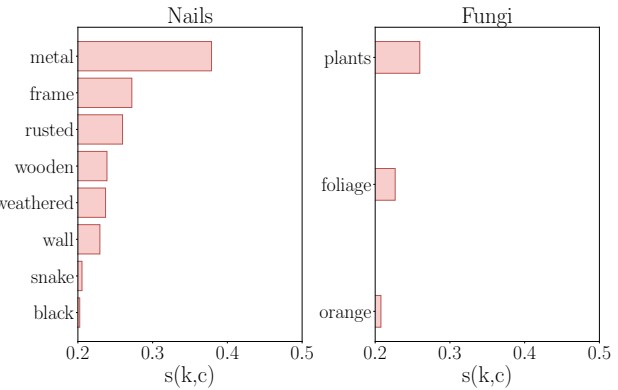

Figure 8: Similarity scores for the *nails* and *fungi* classes from ImageNet-A.

task harder. In this case, we deduce that the model simply was unable to properly fit the two classes because of a lack of samples in the training set.

### 5.4 Human Feedback

To further validate whether the biases found by SaMyNa are meaningful, we have performed a human annotation study on CelebA, Waterbirds, and BAR, by surveying 20 participants. In our survey, we show participants a set of images from each dataset, divided by target class (for example, blond and non-blond people, waterbirds and landbirds, and so on). We asked participants to provide sets of keywords that, in their opinion, represent a bias in the different groups. Then, we analyzed the keywords found by participants and we ranked them based on the number of occurrences. We report the results in Tab. 3 (for CelebA), Tab. 4 (for waterbirds), and Tab. 5 (for BAR). We report the top-5 results on SaMyNa, and all the keywords provided by the participants, which occurred at least three times.

As we can observe from the results, the output of SaMyNa is aligned with human annotations. For more information about the survey, see Sec. E in the Appendix.

| Blond (Human) | | Blond (SaMyNa) | | Not blond (Human) | | Not blond (SaMyNa) | |
|---|---|---|---|---|---|---|---|
| Keyword | Count | Keyword | Score | Keyword | Count | Keyword | Score |
| woman | 20 | woman | 0.40 | man | 14 | man | 0.46 |
| long hair | 5 | blonde | 0.34 | hat | 5 | male | 0.39 |
| white | 4 | mascara | 0.25 | short hair | 4 | | |
| white skin | 4 | makeup | 0.25 | | | | |
| smile | 3 | lipstick | 0.22 | | | | |

Table 3: Comparison between human annotations and SaMyNa's output on the CelebA dataset.

### 5.5 Ablation on $k$

We present here the ablation on $k$, which selects the cardinality of aligned and conflicting samples per class. Fig. 9 reports the study for the most occurring keywords in the cases under exam, while the full results are later reported in Tab. 28 in the Appendix. In the general case, we observe that for lower values of $k$ the similarity score is in general lower, evidencing that the information extraction process is less accurate due to

| Landbird (Human) | | Landbird (SaMyNa) | | Waterbird (Human) | | Waterbird (SaMyNa) | |
|---|---|---|---|---|---|---|---|
| Keyword | Count | Keyword | Score | Keyword | Count | Keyword | Score |
| forest | 6 | forest | 0.33 | water | 5 | ocean | 0.54 |
| trees | 6 | foliage | 0.33 | grey | 4 | beach | 0.43 |
| green | 3 | tree | 0.33 | red | 3 | shore | 0.39 |
| | | stalks | 0.31 | | | water | 0.39 |
| | | branch | 0.31 | | | waves | 0.33 |

Table 4: Comparison between human annotations and SaMyNa's output on the Waterbirds dataset.

| Climbing (Human) | | Climbing (SaMyNa) | | Diving (Human) | | Diving (SaMyNa) | | Fishing (Human) | | Fishing (SaMyNa) | |
|---|---|---|---|---|---|---|---|---|---|---|---|
| Keyword | Count | Keyword | Score | Keyword | Count | Keyword | Score | Keyword | Count | Keyword | Score |
| rocks | 9 | climber | 0.56 | water | 10 | scuba | 0.58 | water | 7 | boat | 0. |
| mountain | 6 | cliff | 0.55 | blue | 5 | underwater | 0.56 | fishing rod | 4 | fishing | 0. |
| helmet | 6 | climbing | 0.52 | sea | 4 | diver | 0.49 | children | 4 | river | 0. |
| rock | 4 | climb | 0.50 | pool | 4 | diving | 0.35 | fish | 4 | ocean | 0. |
| ice | 4 | rock | 0.42 | man | 3 | dive | 0.33 | lake | 3 | lake | 0. |
| Racing (Human) | | Racing (SaMyNa) | | Throwing (Human) | | Throwing (SaMyNa) | | Vaulting (Human) | | Vaulting (SaMyNa) | |
| Keyword | Count | Keyword | Score | Keyword | Count | Keyword | Score | Keyword | Count | Keyword | Score |
| cars | 5 | racing | 0.42 | man | 11 | pitch | 0.55 | sky | 5 | vaulter | 0.58 |
| car | 5 | cars | 0.42 | baseball | 8 | baseball | 0.48 | pole | 4 | vaulting | 0.51 |
| wheels | 5 | car | 0.39 | ball | 4 | pitcher | 0.48 | air | 4 | vault | 0.47 |
| road | 5 | track | 0.32 | sport | 4 | batter | 0.46 | woman | 4 | jump | 0.43 |
| | | stadium | 0.29 | | | mound | 0.44 | | | midair | 0.38 |

Table 5: Comparison between human annotations and SaMyNa's output on the BAR dataset.

a general lack of information (and variety). This trend is particularly evident for $k = 1$. Overall, we find that a fair compromise between performance and complexity is given by the intermediate $k = 10$ for which the scores of bias-related keywords reach a high value with diminishing returns for higher values of $k$. We highlight that maintaining $k$ at bay reduces the number of captions that need to be generated, which is a time-consuming process when using large captioning models. Furthermore, the number of comparisons grows quadratically with $k$, but this is usually not a problem since the captioning process is much slower when $k$ is a reasonable value (for $k \leq 50$ captioning is orders of magnitude slower than calculating the learned class embeddings).

### 5.5.1 Other Ablation Studies

We provide other ablation studies in Sec. C of the Appendix. In particular, we show an ablation on the bias mining step, an ablation on Eq. 4, an experiment with different text embedders, and ablations on all hyperparameters.

### 5.5.2 Other use cases for SaMyNa

In the Appendix, we demonstrate how two additional SaMyNa features can further facilitate keyword interpretation for the end-user. We provide experimental techniques in Sections F and G that further automate this process.

### 5.6 Limitations and Biases from Pretraining

Our method assumes the usage of pre-trained captioning models to extract descriptions (and eventually keywords), potentially representing a bias in the model under analysis. Even if we validated our method both by critically evaluating SaMyNa's outputs and by designing a dedicated human evaluation study to verify that human subjects' interpretations align with the biases highlighted by our algorithm, we recognize that

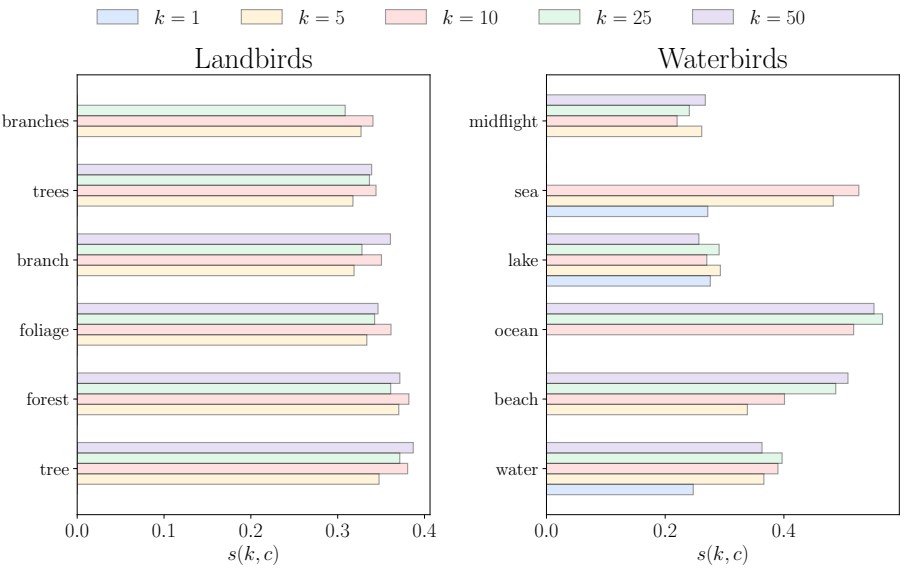

Figure 9: Ablation study on Waterbirds, where we analyze the impact of $k$ that selects the cohort of images to extract keywords from.

our analyses cannot globally prevent the interference of unknown and undesired biases embodied in cases outside the ones we addressed in this paper. Pretrained captioning models (as every deep learning model over which direct control over data and training protocols is not available) may present biases themselves and, as such, exaggerate the presence of certain attributes or completely overlook them. While our results allow us to confidently exclude this for the examined benchmark datasets and employed models, this may be different for other use cases. Accordingly, it's worth underlining that we propose SaMyNa as a supporting tool for human users, who hold the final decision on how to interpret the algorithm's output. This is crucial, as further research is needed to fully understand and prevent the presence of biases in deep learning models, a challenge shared by the whole community.

## 6 Conclusion

In this work, we presented "Say My Name" (SaMyNa), a tool designed to identify and address biases within deep learning models semantically. Unlike similar methods that generate bias pseudo-labels without clear semantic information, SaMyNa offers a text-based pipeline that enhances the explainability of the bias extraction process from the model.
Our approach, validated on well-known benchmarks, proved its effectiveness by both providing the keywords encoding the bias and assigning an interpretable score telling how much the model under analysis is biased to the found attribute. SaMyNa proposes itself not only as a post-hoc analysis tool: through its bias mining approach, it can determine the specific moment the model might be fitting a bias, for which there is in principle no need for a validation set to mine and name the bias. SaMyNa's ambition is to self-establish as a foundational tool in making deep learning models more transparent and fair, offering practical solutions for the scientific community and end-users alike.

### Broader Impact Statement

This work has the potential for a positive societal impact by promoting fairness in AI systems. SaMyNa is designed to uncover and analyze biases in models, providing a clearer understanding of their decision-making processes. By identifying and representing biases in a semantic way, our approach, combined with state-of-the-art debiasing methods, can help mitigate unfairness and support the development of more transparent and equitable machine learning models.

## Acknowledgements

This work was supported in part by the French National Research Agency (ANR) in the framework of the JCJC project "BANERA" under Grant ANR-24-CE23-4369, and in part by the Hi!PARIS Center on Data Analytics and Artificial Intelligence. We acknowledge ISCRA for awarding this project access to the LEONARDO supercomputer, owned by the EuroHPC Joint Undertaking, hosted by CINECA (Italy).

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

# Appendix

## A    Visual feedback

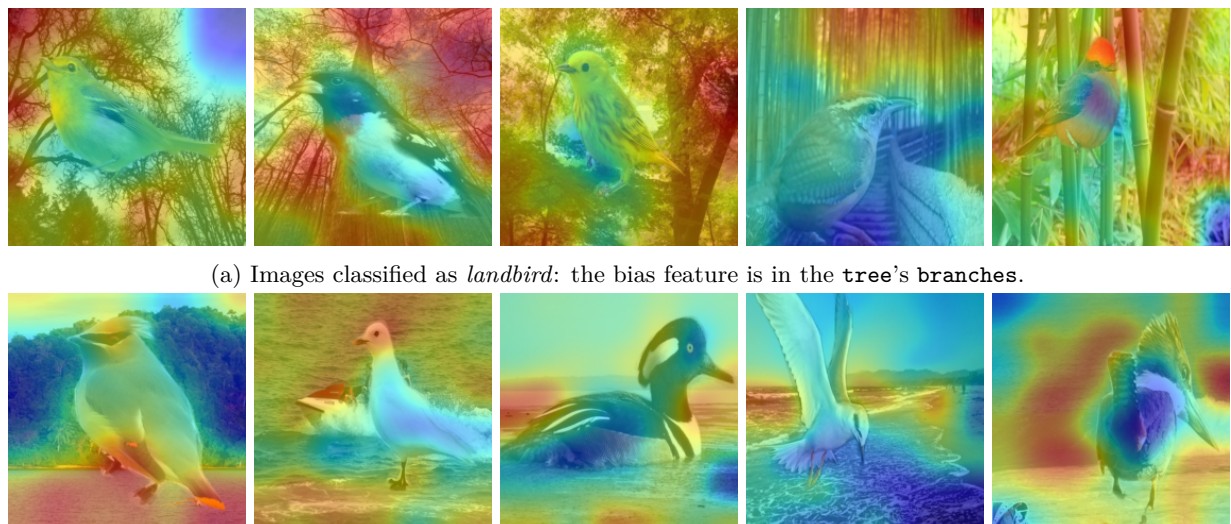

(a) Images classified as *landbird*: the bias feature is in the `tree`'s `branches`.

(b) Images classified as *waterbird*: the bias feature is the `sea`.

Figure 10: Bias heatmaps generated with SaMyNa using CLIP's vision encoder Radford et al. (2021) on Waterbirds.

For visualization purposes only, SaMyNa can leverage a visual encoder instead of a text encoder to identify the part of the image where the potential bias is located. To do this, we adopt the same strategy described in Sec. 3.2.4 to generate the learned class embeddings $E^*(c)$ using image embeddings generated with CLIP's visual encoder Radford et al. (2021)[4] instead of caption embeddings. We can then compute the cosine similarity between $E^*(c)$ and the embeddings of patches from the image we want to analyze. We then plot the similarity values as a heatmap overlay. Fig. 10a and Fig. 10b highlight (in red) that the most salient feature is the tree for the landbird, while it is the sea for the waterbird: the model under exam does not focus on the birds but rather on the background, coherently with what we have observed in Sec. 4.2.1.

## B    Comparison of different captioners

We perform a comparison between different captioners, notably with ClipCap (Mokady et al., 2021) as it has been recently used by related works (Kim et al., 2024). First of all, we are interested in evaluating whether ClipCap can be used to detect biases on CelebA, and how it compares to LLaVA-34b. For this, we report in Tab. 6 the percentage of captions that contain keywords related to CelebA's attributes. As can be seen, ClipCap detects these concepts very rarely, except for "man", which is detected in 10% of the images. LLaVA-34b, on the other hand, detects these keywords much more frequently, for example it detects "man" in 22% of the images, while "woman" is detected in 71.5% of the images (totaling 93.5% detection for gender related keyword, compared to ClipCap's 11.47%). This is a consequence of ClipCap's short captions, that cannot capture enough information from the images. ClipCap's captions are 9 words long on average, while LLaVA's are 137 words long on average. Finally, we show a qualitative comparison of ClipCap's and LaVA's captions in Tab. 7 and Tab. 8.

---

[4]https://huggingface.co/sentence-transformers/clip-ViT-L-14

| Keywords | ClipCap | LLaVA 34b |
|---|---|---|
| old/oldest | 0.04% | 2.00% |
| middle-age/middle-aged | 0.00% | 4.50% |
| young/youngest | 0.86% | 10.0% |
| blond/blonde | 0.02% | 49.0% |
| smile/smiling/smiles | 1.89% | 58.0% |
| tie | 0.05% | 2.00% |
| eyeglasses/glasses/sunglasses | 0.69% | 3.00% |
| beard | 1.70% | 4.50% |
| mustache | 0.25% | 3.50% |
| makeup | 1.27% | 61.0% |
| man/male | 9.26% | 22.0% |
| woman/female | 2.21% | 71.5% |
| hat/hats | 0.84% | 2.00% |
| earring/earrings | 0.02% | 14.5% |
| necklace | 0.02% | 8.50% |

Table 6: The table shows the percentage of captions that contain keywords related to CelebA attributes, both for ClipCap and LLaVA-34B. As can be seen, LLaVA-34B is able to capture much more attributes from CelebA's images. Thus, ClipCap is not suitable for detecting biases other than "man" on CelebA.

| Image | ClipCap | LLaVA-34B |
|---|---|---|
|  | actor with a weight of kg and a feet size of 9 in favorite outfit & clothing style. | The image is a close-up portrait of a woman with fair skin and light-colored hair. Her hair is styled in a side-swept manner, with some strands falling over her forehead. She has a neutral expression on her face and is looking directly at the camera. The woman is wearing makeup, including eyeliner and lipstick, which accentuates her features. She has a small earring visible in her left ear. The background of the image is plain and does not provide any additional context about the location or setting. The lighting on the subject is soft and even, highlighting her facial features without creating harsh shadows. The style of the image suggests it could be from a red carpet event or a formal photoshoot, given the subject's attire and the professional quality of the photograph. |
|  | actor was a popular actor and comedian. | The image is a portrait-style photograph of a man. He appears to be middle-aged with short, dark hair and a fair complexion. The man is wearing a patterned shirt with horizontal stripes in various colors, including red, white, and black. He has a slight smile and is looking directly at the camera. The background is a solid, light blue color, providing a neutral backdrop that contrasts with the subject. There are no visible texts or distinguishing marks on the image that provide additional context or information about the location or the identity of the man. The style of the image is straightforward and seems to be a standard headshot or portrait commonly used for identification or promotional purposes. |

Table 7: Comparison of captions generated by ClipCap and LLaVA-34B on CelebA. The images to caption were selected randomly.

| Image | ClipCap | LLaVA-34B |
|---|---|---|
|  | actor is the face of beauty. | The image is a close-up portrait of a woman with a soft, natural appearance. She has long, wavy blonde hair that falls over her shoulders. Her facial features are symmetrical, with a straight nose, full lips, and clear skin. Her eyes are a light color, possibly blue or green, and she is looking directly at the camera with a gentle expression. The woman's makeup is subtle, enhancing her natural beauty without appearing overly dramatic. The background of the image is a solid, neutral color, providing a clean and uncluttered backdrop that allows the focus to remain on the woman. The lighting in the photograph is soft and even, which contributes to the gentle and approachable look of the subject. There are no visible texts or logos in the image, and the style of the photograph suggests it could be used for a variety of purposes, such as a professional headshot, a beauty advertisement, or a personal portrait. The image does not provide any additional context about the location or setting, as the focus is solely on the woman's face and upper body. |
|  | actor with her hair in a bob. | The image is a close-up portrait of a woman with a smiling expression. She has short, dark hair and is wearing makeup that includes dark eyeliner and eyeshadow, as well as what appears to be a light foundation or powder on her face. Her lips are slightly parted, and she is looking directly at the camera. The woman is wearing a black garment that is not fully visible in the frame. The background is blurred, but it suggests an indoor setting with architectural features such as arches and what might be a stone or brick wall. The lighting on the subject is bright, highlighting her features and the contours of her face. The style of the image is a standard portrait with a focus on the subject's face and expression. There are no visible texts or logos in the image. |

Table 8: Comparison of captions generated by ClipCap and LLaVA-34B on CelebA. The images to caption were selected randomly.

# C Ablation Study

## C.1 Ablations on the Bias Mining step

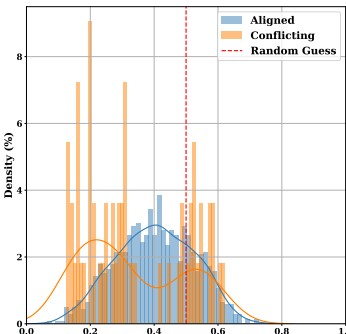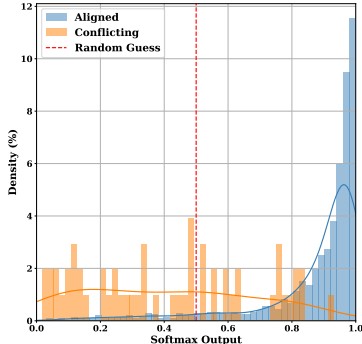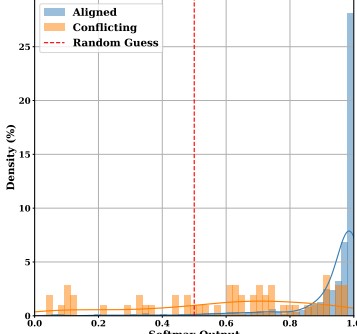

Figure 11: Output distributions *Waterbirds* target class from the ResNet-50 trained on Waterbirds at early stages (epoch 1, left), at extraction time $t^*$ (epoch 6, center) and in the final stage (epoch 10, right).

In this section, we provide visualizations on the output distributions on the *Waterbirds* target when training a ResNet-50 on Waterbirds in the same setup as described in Sec. 4.1. Fig. 11 proposes visualizations of the output distributions for the bias-target aligned samples in blue (*waterbirds* and sea landscape), and bias-target conflicting samples in orange (waterbirds and ground landscape), in three different moments of the training: in the early stages (at $t = 1$, on the left, where $t$ is the training epoch), at the chosen bias extraction time $t^*$ ($t = 6$, at the center) and in the final stages ($t = 10$, on the right). We recall that the entire bias extraction process happens directly on the training set $\mathcal{D}^{\text{train}}$ and does not require the employment of a validation/test set. We observe here that, at extraction time, there is an evident separation between bias-aligned and bias-conflicting samples, and we are able to confidently isolate the most biased among the conflicting samples (the orange distribution having a larger population below the random guess threshold). This does not hold in case the extraction time is delayed, with the bias-conflicting distribution not exhibiting a peak anymore.

## C.2 Ablation on Bias Naming

### C.2.1 Ablation of Eq. 4

To demonstrate the effectiveness of Eq. 4 in filtering out the shared information across all dataset we did an ablation study on the CelebA dataset. Fig. 12 shows the top-9 keywords for each class on CelebA when Eq. 4 is removed from our method. As can be seen, the ranking is full of irrelevant keywords that can be used to describe just about any image in the dataset, like `portrait`, `photo`, `image`, and `headshot`.

### C.2.2 Ablation on Text Embedders

In this subsection, we are interested in testing SaMyNa with more diverse models for the textual embedding, in an attempt to check the generality of the proposed approach. We provide, in Tab. 9, the results obtained with five other popular textual embedding models. From our results, we can clearly see that when employing any of the tested models, we are able to find back the two typical biases from waterbirds. All models produce similar outputs, except for DistilRoberta and NOMIC, that produce very few, albeit correct, keywords for Landbirds.

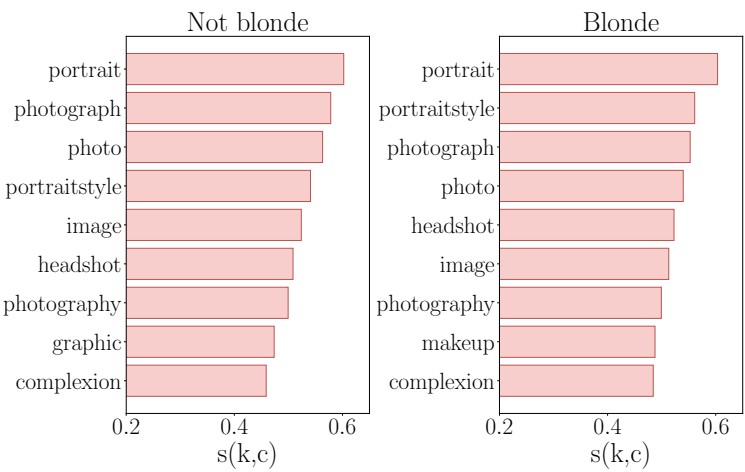

Figure 12: Similarity scores for the CelebA dataset when ablating Eq. 4 from our method.

| Embedding Model | Target | Keyword (value) |
|---|---|---|
| DistilRoberta | *Landbirds* | tree (0.22477) |
| | *Waterbirds* | ocean (0.41353), shore (0.37904), beach (0.37184), boat (0.35631), water (0.28810), river (0.28312), waves (0.28209), sandy (0.25046), seagull (0.23950), lake (0.22972), across (0.21785), foam (0.20274) |
| All-MiniLM-L12-v2 | *Landbirds* | tree (0.35688), forest (0.33959), trees (0.31249), foliage (0.27917), branches (0.25548), vegetation (0.25477), branch (0.25477), leaves (0.21215) |
| | *Waterbirds* | ocean (0.40874), boat (0.35452), shore (0.31332), waves (0.27757), beach (0.27478), seagull (0.23625), lake (0.22088) |
| MPNET | *Landbirds* | tree (0.38503), forest (0.31970), trees (0.31858), foliage (0.26440), branches (0.25911), branch (0.23637), bird (0.22693), bamboo (0.22661), perched (0.20324) |
| | *Waterbirds* | waves (0.34030), ocean (0.33423), beach (0.31522), shore (0.26980), water (0.22304) |
| NOMIC | *Landbirds* | forest (0.22448), branches (0.21085) |
| | *Waterbirds* | waves (0.30429), ocean (0.29857), beach (0.26858), water (0.23373), boat (0.21911), shore (0.21033), lake (0.20263) |
| AlBERT | *Landbirds* | trees (0.32898), tree (0.32710), foliage (0.31355), forest (0.30978), branch (0.29175), branches (0.29141), vegetation (0.28953), bamboo (0.27675), stalks (0.23657) |
| | *Waterbirds* | ocean (0.49492), beach (0.47047), waves (0.42517), water (0.38388), shore (0.35404), boat (0.29732), foam (0.26902), lake (0.26790), sandy (0.25054), surface (0.24005), midflight (0.21686), river (0.21444) |

Table 9: Ablation study on Waterbirds, where we analyze the impact of employing a diverse encoder for SaMyNa.

### C.2.3 Ablation on $f_{\min}$

We propose here, in Table 10, the results we obtain by varying the value of the hyperparameter $f_{\min}$, i.e. the minimum fraction of captions from examples of the same predicted class in which a keyword must appear so that it can be considered as a possible output. As $f_{\min}$ increases, each new set of selected keywords is a

subset of the previous one, with all keywords maintaining identical scores. When $f_{\min}$ is too high, one class will be wrongfully marked as not holding a bias (with $f_{\min} = 0.60$). On the other hand, when not employing filtering at all ($f_{\min} = 0$), a lot of more fine-grained classes like `coniferous` or `wilderness` arise. We chose a low threshold ($f_{\min} = 0.15$) to filter out just the least frequent words that may come from mistakes of the captioner or from details that appear in just a few images. Filtering out too many words with this method may remove important synonyms from the ranking, so we avoid it.

$f_{\min} = 0.0$

| Landbirds | | Waterbirds | |
|---|---|---|---|
| shrubs | 0.35 | sea | 0.54 |
| forest | 0.33 | ocean | 0.54 |
| foliage | 0.33 | aquatic | 0.44 |
| twigs | 0.33 | beach | 0.43 |
| tree | 0.33 | harbor | 0.40 |
| deciduous | 0.32 | ships | 0.39 |
| stalks | 0.31 | shore | 0.39 |
| branch | 0.31 | water | 0.39 |
| forested | 0.31 | shoreline | 0.38 |
| trees | 0.30 | pier | 0.37 |
| branches | 0.30 | ship | 0.37 |
| bushes | 0.30 | coastal | 0.35 |
| wooded | 0.29 | waves | 0.33 |
| vegetation | 0.27 | sailboat | 0.32 |
| weeds | 0.26 | coastline | 0.32 |
| grasses | 0.24 | sail | 0.30 |
| greenery | 0.23 | wave | 0.30 |
| wilderness | 0.22 | vessel | 0.30 |
| stalk | 0.22 | floating | 0.29 |
| coniferous | 0.22 | boat | 0.29 |
| garden | 0.21 | speedboat | 0.29 |
| bark | 0.20 | wetsuit | 0.29 |
| grayishbrown | 0.20 | pond | 0.29 |
| | | swimming | 0.28 |
| | | lagoon | 0.28 |
| | | bay | 0.27 |
| | | splash | 0.27 |
| | | river | 0.27 |
| | | surfboard | 0.27 |
| | | wake | 0.26 |
| | | surfer | 0.26 |
| | | glides | 0.25 |
| | | surfing | 0.25 |
| | | dock | 0.25 |
| | | pool | 0.25 |
| | | midflight | 0.24 |
| | | lake | 0.24 |
| | | cranes | 0.22 |
| | | flying | 0.22 |
| | | sand | 0.21 |
| | | watermark | 0.20 |

$f_{\min} = 0.1$

| Landbirds | | Waterbirds | |
|---|---|---|---|
| shrubs | 0.35 | ocean | 0.54 |
| forest | 0.33 | beach | 0.43 |
| foliage | 0.33 | shore | 0.39 |
| tree | 0.33 | water | 0.39 |
| deciduous | 0.32 | shoreline | 0.38 |
| stalks | 0.31 | coastal | 0.35 |
| branch | 0.31 | waves | 0.33 |
| trees | 0.30 | wave | 0.30 |
| branches | 0.30 | boat | 0.29 |
| vegetation | 0.27 | river | 0.27 |
| grasses | 0.24 | midflight | 0.24 |
| | | lake | 0.24 |
| | | flying | 0.22 |
| | | watermark | 0.20 |

$f_{\min} = 0.15$

| Landbirds | | Waterbirds | |
|---|---|---|---|
| forest | 0.33 | ocean | 0.54 |
| foliage | 0.33 | beach | 0.43 |
| tree | 0.33 | shore | 0.39 |
| stalks | 0.31 | water | 0.39 |
| branch | 0.31 | waves | 0.33 |
| trees | 0.30 | boat | 0.29 |
| branches | 0.30 | river | 0.27 |
| vegetation | 0.27 | midflight | 0.24 |
| | | lake | 0.24 |
| | | flying | 0.22 |

$f_{\min} = 0.30$

| Landbirds | | Waterbirds | |
|---|---|---|---|
| forest | 0.33 | ocean | 0.54 |
| branch | 0.31 | water | 0.39 |
| trees | 0.30 | waves | 0.33 |

$f_{\min} = 0.45$

| Landbirds | | Waterbirds | |
|---|---|---|---|
| trees | 0.30 | water | 0.39 |

$f_{\min} = 0.60$

| Landbirds | Waterbirds | |
|---|---|---|
| | water | 0.39 |

Table 10: Ablation study on Waterbirds, where we analyze the impact of the threshold for the frequency for the found keywords ($f_{\min}$).

### C.2.4  Ablation on $t_{sim}$

In this ablation study, we provide an example of the keywords resulting from SaMyNa when not employing any thresholding on the minimum similarity values for the found keywords. In Table 11, we provide the full

output only for *landbirds*, due to the excessive length of the unfiltered output. However, this is not an issue because SaMyNa, when applied to binary classification, possesses a mathematical property that guarantees symmetrical keyword rankings for the two classes. Consequently, the keyword ranking for *waterbirds* is simply the inverse of the ranking for *landbirds* (the cosine similarity scores are the same but with opposite sign). Nevertheless, we provide the full list of the resulting keywords as a text file, available in the supplementary materials zip archive (`keywords_ablation_tsim.txt`). From an analysis of the emerging keywords, we observe three interesting intervals of values. High similarity values indicate those concepts that correlate well specifically with the learned class and are those presented in the paper. Concepts whose similarity is close to zero are not correlated: indeed, we can find keywords like `possibly`, `various`, and `open` that are neutral concepts. Interestingly, we can also identify concepts like `background` and `environment` that are super-classes of the two biases. This confirms that SaMyNa works properly since it puts itself in the best spot to best discriminate the two biases. Finally, the third region is for negatively correlated concepts (anti-correlated), where we easily find the concepts correlated with the other class (*waterbirds*) given that we are in a binary classification task.

### C.2.5  Ablation on $f_{min}$, $t_{sim}$, and k

In this ablation study, we show the compound effect of the $f_{min}$, $t_{sim}$, and $K$ hyperparameters. By setting $f_{min} = 0$ and $t_{sim} = -1$, we effectively disable filtering. We also set $k = 50$ since it is the maximum value of $k$ we tried, and will produce more captions, and, by extension, more keywords. Tab. 12 shows the results for both classes of Waterbirds. Due to space constraints, we show only the top keywords. The total number of keywords ranked for each class is 1537 and the full result can be consulted in the supplementary materials zip archive (`keywords_ablation_all_hyperparams.txt`).

| Use of $t_{\text{sim}}$ | Keyword (value) |
|---|---|
| $t_{\text{sim}} = -1$ (*Landbirds*) | forest (0.33454), foliage (0.33412), tree (0.32720), stalks (0.31389), branch (0.31242), trees (0.29919), branches (0.29649), vegetation (0.27349), leaves (0.19398), bamboo (0.16775), brown (0.16085), small (0.15988), black (0.15947), stands (0.15171), green (0.15048), gray (0.14847), nature (0.14642), yellow (0.14465), perched (0.14381), habitat (0.14101), positioned (0.12865), corner (0.12602), position (0.12585), style (0.12585), standing (0.12558), color (0.12359), bird (0.12310), natural (0.12179), webbed (0.11827), naturalistic (0.11359), lush (0.11342), darker (0.11082), dark (0.10989), colors (0.10947), features (0.10804), slightly (0.10770), texts (0.10516), photography (0.10478), markings (0.10345), lighting (0.10088), plumage (0.09747), shades (0.09703), setting (0.09479), camera (0.08995), slender (0.08957), soft (0.08622), feathers (0.08274), beak (0.08256), pointed (0.08064), facing (0.08016), center (0.07854), feet (0.07604), area (0.06895), vibrant (0.06787), short (0.06720), frame (0.06472), depicts (0.06152), appears (0.06135), central (0.06021), background (0.06013), ground (0.05912), creating (0.05908), photograph (0.05901), elements (0.05896), side (0.05860), located (0.05815), main (0.05688), subject (0.05661), top (0.05637), snapshot (0.05501), looking (0.05431), marks (0.05423), focal (0.05365), eyes (0.05237), suggests (0.05027), composite (0.04975), eye (0.04772), lighter (0.04710), presence (0.04675), create (0.04601), tall (0.04574), turned (0.04134), perspective (0.03960), indicating (0.03954), calm (0.03905), predominantly (0.03879), suggesting (0.03870), path (0.03797), thin (0.03732), light (0.03667), moment (0.03628), left (0.03577), touch (0.03549), surrounding (0.03547), wide (0.03536), lower (0.03233), featuring (0.03144), distinctive (0.03032), landscape (0.02956), onto (0.02917), red (0.02857), consists (0.02824), foreground (0.02821), bottom (0.02585), scale (0.02508), sense (0.02173), context (0.02116), focus (0.02088), possibly (0.02039), capturing (0.01853), around (0.01757), legs (0.01733), visible (0.01702), day (0.01428), visually (0.01370), composition (0.01359), point (0.01284), scene (0.01166), might (0.01119), making (0.01009), blue (0.01004), distinguishing (0.01002), reflecting (0.00721), tail (0.00713), amidst (0.00643), open (0.00581), objects (0.00573), providing (0.00562), quality (0.00505), towards (0.00415), image (0.00329), taking (0.00250), captured (0.00037), buildings (-0.00146), viewer (-0.00204), peaceful (-0.00231), taken (-0.00423), wings (-0.00644), various (-0.00649), environment (-0.00728), movement (-0.00924), view (-0.00953), head (-0.01000), scattered (-0.01045), provide (-0.01047), overall (-0.01051), fully (-0.01158), detail (-0.01264), captures (-0.01321), near (-0.01394), likely (-0.01408), tranquil (-0.01455), appealing (-0.01471), drawing (-0.01530), body (-0.01555), expanse (-0.01751), could (-0.01791), backdrop (-0.01856), large (-0.02118), immediately (-0.02224), white (-0.02302), right (-0.02431), adding (-0.02552), realistic (-0.03162), deep (-0.03255), surroundings (-0.03269), otherwise (-0.03385), attention (-0.03709), beautifully (-0.03766), relative (-0.03802), beautiful (-0.03860), sunny (-0.04031), tranquility (-0.04087), freedom (-0.04306), contrast (-0.04520), also (-0.04593), sky (-0.04655), across (-0.04660), blend (-0.04802), extended (-0.04905), depth (-0.05085), juxtaposes (-0.05146), dense (-0.05250), adds (-0.06059), two (-0.07112), clouds (-0.07306), dynamic (-0.07317), spread (-0.07693), contrasting (-0.07756), surface (-0.07825), harmonious (-0.08196), serene (-0.08365), world (-0.10627), distance (-0.10900), moving (-0.11027), mix (-0.11326), filled (-0.12270), soars (-0.13097), overcast (-0.14479), flight (-0.14884), sandy (-0.14945), clear (-0.16600), seagull (-0.18060), foam (-0.19652), flying (-0.21832), lake (-0.24355), midflight (-0.24404), river (-0.26848), boat (-0.29352), waves (-0.33462), water (-0.38571), shore (-0.38639), beach (-0.43417), ocean (-0.53961) |

Table 11: Ablation study on Waterbirds, where we analyze the impact of the threshold for the similarity score ($t_{\text{sim}}$) for the class *landbirds*.

| Class | Keyword (value) |
|---|---|
| Landbirds | tree (0.35786), forest (0.35347), shrubbery (0.34675), shrubs (0.34670), foliage (0.34207), leaf (0.33083), twigs (0.32846), deciduous (0.32429), trees (0.32224), stalks (0.31635), branch (0.31585), plants (0.31248), forested (0.31134), plant (0.30522), bushes (0.29749), wooded (0.29437), branches (0.29422), planted (0.27602), woodpecker (0.26986), woods (0.26887), vegetation (0.26780), pine (0.26381), weeds (0.25234), leaves (0.23516), grasses (0.23446), flora (0.22909), owl (0.22551), twig (0.22138), garden (0.21504), bark (0.21230), stalk (0.20237), greenishbrown (0.20123), coniferous (0.20117), grayishbrown (0.19607), greenishgray (0.19218), wood (0.18801), greenery (0.18465), wilderness (0.18064), lily (0.17892), yellowishgreen (0.17775), greenish (0.16826), yellowishbrown (0.16707), porch (0.16707), stripe (0.16603), bamboo (0.16482), reddishbrown (0.16392), trunk (0.16291), fence (0.16257), moss (0.15525), grayishblack (0.15461), flower (0.15428), brown (0.14858), claws (0.14673), songbird (0.14653), greenishblue (0.14383), squirrels (0.14337), grayish (0.14330), trail (0.14072), green (0.13819), wooden (0.13784), striped (0.13633), grass (0.13416), sparrow (0.13305), nest (0.13205), pouch (0.12906), nesting (0.12292), smokestacks (0.12155), canopy (0.11981), species (0.11944), patterned (0.11719), gray (0.11595), grey (0.11442), flowers (0.11391), darkcolored (0.11316), bluebird (0.11249), curved (0.11216), row (0.10993), cracks (0.10955), plaid (0.10909), wires (0.10906), line (0.10826), black (0.10768), cutting (0.10540), potted (0.10506), border (0.10473), crouched (0.10454), foot (0.10222), pole (0.10100), bird (0.10011), handlebars (0.09951), reeds (0.09944), bordered (0.09866), standup (0.09824), gannet (0.09747), yellow (0.09704), crow (0.09696), shaded (0.09692), beige (0.09642), neck (0.09496), eagle (0.09413), corner (0.09216), gate (0.09152), blade (0.09038), blues (0.09002), ring (0.08994), lined (0.08993) |
| Waterbirds | sea (0.58878), ocean (0.57980), beach (0.53000), beachgoers (0.48649), tide (0.48087), shore (0.47279), shoreline (0.46738), aquatic (0.46162), coastal (0.44221), harbor (0.43874), maritime (0.42833), pier (0.41840), coastline (0.40540), marina (0.40293), submarine (0.40068), water (0.40000), ships (0.39669), ship (0.37869), waterfront (0.36526), waves (0.36266), boats (0.35816), beachwear (0.35474), lagoon (0.35376), bikini (0.34825), submerged (0.34601), naval (0.33164), floating (0.33147), pond (0.32841), wave (0.32569), bay (0.32533), wet (0.32244), wetsuit (0.31642), swim (0.31542), sailboat (0.31461), crab (0.31139), boat (0.31050), sail (0.30284), wake (0.30191), bathing (0.30030), swimming (0.29998), sand (0.29526), lake (0.29238), fish (0.29191), river (0.28856), vessel (0.28363), dock (0.27980), surfboard (0.27760), pool (0.27661), splash (0.26832), sinking (0.26343), liquid (0.26182), watermark (0.26116), surfer (0.26046), midflight (0.25716), speedboat (0.25574), coral (0.24456), cranes (0.24285), docked (0.24119), surfing (0.24079), flows (0.23242), jetty (0.22524), flying (0.22223), sandy (0.21984), island (0.21959), lakeside (0.21866), glides (0.21866), diving (0.21596), foam (0.21396), skyline (0.20477), horizon (0.19709), seagull (0.19641), mirror (0.19333), mirrorlike (0.19307), stunning (0.19107), mirrors (0.18967), inland (0.18922), pontvie (0.18885), soaring (0.18690), picturesque (0.18462), dive (0.18419), meal (0.18359), serene (0.18255), paddleboarder (0.17829), waterfall (0.17474), flowing (0.17394), cloud (0.17373), panoramic (0.17271), paddleboard (0.16915), paddleboarding (0.16890), rocky (0.16798), clear (0.16612), gliding (0.16549), surface (0.16489), ripples (0.16461), sunset (0.16234), wading (0.16154), bathed (0.15626), waterfowl (0.15250), multiple (0.15204), overcast (0.15028), whimsy (0.14977), paddle (0.14693), stream (0.14617), buoy (0.14510), dam (0.14358), reality (0.14326) |

Table 12: Ablation study on Waterbirds, where we analyze simultaneously the impact of $t_{\mathrm{sim}}$, $f_{\mathrm{min}}$, and $K$. Filtering is disabled by setting $t_{\mathrm{sim}} = -1$ and $f_{\mathrm{min}} = 0$. We use $k = 50$ to generate more captions and, as a consequence, more keywords.

| Method | Captioning Time (LLaVA 34B) |
|---|---|
| SaMyNa (k=1) | 17 minutes |
| SaMyNa (k=5) | 86 minutes |
| SaMyNa (k=10) | 3 hours |
| SaMyNa (k=25) | 7 hours |
| SaMyNa (k=50) | 14 hours |
| B2T | 60 days |

Table 13: Comparison of captioning runtime between SaMyNa (for different values of $k$) and B2T using LLaVA-34B on CelebA. B2T must caption the whole validation set of CelebA, which is composed of about 19k images, while SaMyNa uses bias-mining to select a sample of $k * C * 2$ images, where $C$ is the number of classes. By default, SaMyNa uses $k = 10$, for a total of 40 samples on CelebA.

## D  Comparison with B2T

In this section, we provide a more in-depth comparison with the B2T algorithm proposed by Kim et al. (2024). B2T is relevant to our work, as it shares the same goal of extracting human-interpretable descriptions of potential biases affecting visual models. The key differences between SaMyNa and B2T can be summarized as follows:

- SaMyNa does not require a validation set for bias discovery, in contrast to B2T;

- SaMyNa can extract few candidate exemplars directly from the training set thanks to the Bias Mining step.

This represents a key aspect in unsupervised bias discovery and mitigation, as in a realistic scenario a validation set comprising conflicting samples is rarely available (like in BAR or BFFHQ). Besides, B2T requires captioning the entire validation set in order to extract relevant biases from conflicting samples. In contrast, SaMyNa is much more efficient and allows for the usage of larger and more accurate captioners such as LLaVa-34B (see Sec. B for a comparison between ClipCap and LLaVA-34B).

**Why B2T cannot leverage better captioners while SaMyNa can.**  The quality of extracted keywords directly depends on the quality of the captioner. B2T is forced to employ smaller and quicker captioners such as ClipCap, which, however, provides less accurate captions when compared to models such as LLaVA-34B. Tab. 13 shows the time required for SaMyNa and B2T on the CelebA dataset using LLaVA-34B on an NVIDIA A40 equipped with 48 GB, tested on batch size 5.

**Why B2T requires a full validation set and SaMyNa does not.**  To showcase that B2T requires a large enough validation set to extract accurate keywords, we compare the keywords extracted by SaMyNa and B2T with varying sample sizes. The results are presented in Tab. 14. We highlight cases in which the relevant keyword was ranked higher than the other method. The results clearly show that our method, which leverages the bias mining step, is consistently more accurate than B2T in extracting the right keywords. Furthermore, keep in mind that B2T keywords need to be inverted as reported in the original paper (e.g. woman $\rightarrow$ man), thus for k=1, B2T actually predicts the opposite bias. This is straightforward for a binary attribute such as CelebA's gender, but not obvious when more than two biases are present in the training set. For completeness of results, we report the full ranking of both algorithms for all values of k, in Tab. 15 (k=1), Tab. 16 (k=5), Tab. 17 (k=10), Tab. 18 (k=25), and Tab. 19 (k=50).

**Why B2T cannot use a held-out validation set.**  One could ask why B2T is not applicable if a validation set is missing, because one could generate a held-out validation set from the training set. However, this is not as simple as it seems: B2T on a held-out validation set is not guaranteed to work because of the lack of a bias mining step that ensures that misclassified examples can be found in the validation set for all classes. Without stopping the training at the right epoch, the model will start to generalize and will make

| Class | Method | Expected | K=1 | K=5 | K=10 | K=25 | K=50 |
|---|---|---|---|---|---|---|---|
| **Blond** | **B2T** | **man** | woman (6th) | N/A | N/A | N/A | N/A |
| | **SaMyNa** | **woman** | N/A | woman (6th) | woman (1st) | woman (5th) | woman (1st) |
| **Not Blond** | **B2T** | **woman** | N/A | woman (5th) | woman (3rd) | woman (5th) | woman (4th) |
| | **SaMyNa** | **man** | male (1st) | male (1st) | man (1st) | man (1st) | man (1st) |

Table 14: Comparison between SaMyNa and B2T using the same captioner (LLaVA-34B) and using an equally sized subsample of CelebA's validation set. The number of sample images is $K * 4$. For B2T we selected K random images for the correctly classified, and K for the incorrectly classified examples of each class. For SaMyNa, we use our subsampling algorithm. B2T's expected answer is the opposite of SaMyNa's. We show the position in the ranking of gender keywords. For both algorithms, the default filtering method is used, except for SaMyNa, where we don't filter the target class since it's also not filtered by B2T. As can be seen, SaMyNa detects the bias for "not blond" every time, while for the "not blond" class, it fails only for $K = 1$. B2T, on the other hand, detects the bias for the "not blond" class 4 times out of 5 with worse ranking positions than SaMyNa, while for the "blond" class, it never detects the bias, and for $K = 1$, its answer is the opposite of the expected answer.

| Blond (B2T) | | Blond (SaMyNa) | | Not blond (B2T) | | Not blond (SaMyNa) | |
|---|---|---|---|---|---|---|---|
| Keyword | CLIP Score | Keyword | Cosine Similarity | Keyword | CLIP Score | Keyword | Cosine Similarity |
| wavy | 1.844 | earrings | 0.27081 | blonde hair | 1.594 | male | 0.32069 |
| fair complexion | 1.781 | makeup | 0.26793 | | | grass | 0.24336 |
| close-up | 1.609 | blonde | 0.21713 | | | subject | 0.23409 |
| close-up photograph | 1.5625 | lipstick | 0.20272 | | | environment | 0.21606 |
| neutral expression | 1.375 | eyeshadow | 0.20149 | | | camera | 0.21347 |
| woman | 1.078 | | | | | capturing | 0.21132 |
| complexion | 0.7344 | | | | | mood | 0.20288 |
| lips slightly | 0.4688 | | | | | | |
| complexion and long | 0.375 | | | | | | |
| lips slightly parted | 0.375 | | | | | | |
| hair | 0.2812 | | | | | | |

Table 15: Full keyword rankings of B2T and SaMyNa for the results shown in Tab. 14 with $K = 1$. As can be seen, while SaMyNa fails for the "blond" class, it detects keywords that still make sense like "makeup", "earrings", and "lipstick", which are usually associated with women. While B2T fails in a worse way, since its answer is the polar opposite of the expected one (it should answer male). B2T also fails for the "not blond" class.

fewer classification errors. To show this, we test B2T on BAR, which is a dataset that lacks a validation set. Furthermore, BAR is interesting because it has 6 classes. As previously noted, B2T generates keywords that represent the opposite of the bias, which is easy to interpret in the binary classification case (gender in CelebA), but it loses interpretability in datasets like BAR that have more than 2 classes. For the experiment, we extract a held-out validation split by stratifying with respect to the known class population distributions. The held-out validation set is composed of 195 examples (10% of the training set). The results are shown in Tab. 20. As can be seen, the results for the "Climbing" class are missing because no misclassified examples were found in the held-out validation set, which proves our point. As highlighted in the table's caption, the results for the rest of the classes are mostly wrong, probably due to the low number of misclassified examples. Our method achieves good results, as shown in Tab. 26. Furthermore, we achieved these results without relying on a validation set and by using only 120 samples (K=10) as opposed to the 195 examples we extracted for B2T.

| Blond (B2T) | | Blond (SaMyNa) | | Not blond (B2T) | | Not blond (SaMyNa) | |
|---|---|---|---|---|---|---|---|
| Keyword | CLIP Score | Keyword | Cosine Similarity | Keyword | CLIP Score | Keyword | Cosine Similarity |
| provide additional context | 0.5938 | blonde | 0.38934 | wavy blonde hair | 4.734 | male | 0.30272 |
| provide | 0.5625 | makeup | 0.34895 | wavy blonde | 3.922 | man | 0.27394 |
| distinguishing | 0.5 | mascara | 0.33196 | blonde | 3.719 | shirt | 0.20739 |
| additional context | 0.4844 | lipstick | 0.31817 | blonde hair | 3.64 | | |
| context | 0.4688 | eyeliner | 0.31324 | woman | 1.844 | | |
| texts | 0.3125 | woman | 0.29510 | wearing makeup | 1.3125 | | |
| distinguishing marks | 0.2656 | eyeshadow | 0.24667 | eyeliner | 1.297 | | |
| additional | 0.1562 | shadows | 0.24538 | eyes | 1.125 | | |
| provide additional | 0.1406 | hair | 0.23935 | makeup | 1.078 | | |
| style | 0.03125 | face | 0.22412 | camera | 0.9062 | | |
| | | head | 0.20731 | expression | 0.703 | | |
| | | styled | 0.20134 | long | 0.578 | | |
| | | | | directly | 0.5156 | | |
| | | | | wearing | 0.2188 | | |
| | | | | hair | 0.0 | | |

Table 16: Full keyword rankings of B2T and SaMyNa for the results shown in Tab. 14 with $K = 5$.

| Blond (B2T) | | Blond (SaMyNa) | | Not blond (B2T) | | Not blond (SaMyNa) | |
|---|---|---|---|---|---|---|---|
| Keyword | CLIP Score | Keyword | Cosine Similarity | Keyword | CLIP Score | Keyword | Cosine Similarity |
| eyeliner | 0.9688 | woman | 0.39947 | wavy blonde hair | 3.594 | man | 0.45820 |
| wearing makeup | 0.4844 | blonde | 0.34301 | blonde hair | 2.938 | male | 0.39175 |
| photograph | 0.03125 | mascara | 0.24959 | woman | 1.828 | | |
| | | makeup | 0.24877 | light-colored hair | 1.359 | | |
| | | lipstick | 0.22272 | close-up | 0.01563 | | |
| | | eyeliner | 0.20065 | | | | |

Table 17: Full keyword rankings of B2T and SaMyNa for the results shown in Tab. 14 with $k = 10$.

| Blond (B2T) | | Blond (SaMyNa) | | Not blond (B2T) | | Not blond (SaMyNa) | |
|---|---|---|---|---|---|---|---|
| Keyword | CLIP Score | Keyword | Cosine Similarity | Keyword | CLIP Score | Keyword | Cosine Similarity |
| visible texts | 1.0 | blonde | 0.45956 | wavy blonde hair | 3.766 | man | 0.40512 |
| close-up photograph | 0.7656 | makeup | 0.32295 | blonde | 3.547 | | |
| provide additional context | 0.6406 | mascara | 0.32257 | blonde hair | 3.438 | | |
| directly | 0.5938 | lipstick | 0.30535 | blonde hair styled | 3.438 | | |
| additional context | 0.5156 | woman | 0.30441 | woman | 1.594 | | |
| portrait-style photograph | 0.4062 | eyeliner | 0.24918 | wearing makeup | 0.797 | | |
| fair | 0.4062 | hair | 0.22028 | makeup | 0.672 | | |
| face | 0.1875 | eyeshadow | 0.20621 | fair skin | 0.3906 | | |
| | | | | eyes | 0.2812 | | |
| | | | | hair styled | 0.2344 | | |
| | | | | styled | 0.2344 | | |
| | | | | portrait | 0.2188 | | |

Table 18: Full keyword rankings of B2T and SaMyNa for the results shown in Tab. 14 with $K = 25$.

| Blond (B2T) | | Blond (SaMyNa) | | Not blond (B2T) | | Not blond (SaMyNa) | |
|---|---|---|---|---|---|---|---|
| Keyword | CLIP Score | Keyword | Cosine Similarity | Keyword | CLIP Score | Keyword | Cosine Similarity |
| close-up photograph | 0.8438 | woman | 0.42156 | wavy blonde hair | 4.0 | man | 0.47463 |
| visible texts | 0.797 | blonde | 0.39872 | blonde hair | 3.5 | shirt | 0.20972 |
| photograph | 0.6875 | mascara | 0.29177 | blonde | 3.406 | | |
| expression | 0.5 | lipstick | 0.29163 | woman | 1.375 | | |
| additional context | 0.4531 | makeup | 0.26443 | wearing makeup | 0.7188 | | |
| provide additional context | 0.4375 | | | makeup | 0.5625 | | |
| visible | 0.3594 | | | close-up | 0.1875 | | |
| person | 0.2812 | | | eyeliner | 0.1719 | | |
| texts or distinguishing | 0.2656 | | | fair | 0.1562 | | |
| provide additional | 0.25 | | | hair | 0.0781 | | |
| distinguishing marks | 0.1875 | | | | | | |
| style | 0.1875 | | | | | | |
| wearing | 0.1406 | | | | | | |
| face | 0.03125 | | | | | | |

Table 19: Full keyword rankings of B2T and SaMyNa for the results shown in Tab. 14 with $k = 50$.

| Climbing | | Diving | | Fishing | | Racing | | Throwing | | Vaulting | |
|---|---|---|---|---|---|---|---|---|---|---|---|
| Keyword | CLIP Score | Keyword | CLIP Score | Keyword | CLIP Score | Keyword | CLIP Score | Keyword | CLIP Score | Keyword | CLIP Score |
| empty (crashed) | N/A | soldiers jump | 5.11 | boy fishing | 3.438 | start | 1.656 | friends playing football | 7.67 | empty (all filtered) | N/A |
| | | trampoline | 4.72 | beach | 3.11 | track | 1.594 | american football | 7.188 | | |
| | | swimmers jump | 4.61 | boy | 1.828 | race | 1.484 | american football team | 6.547 | | |
| | | young man jumping | 4.25 | fishing | 0.4375 | motorcycle | 1.219 | playing football | 6.094 | | |
| | | man jumps | 3.969 | man fishing | 0.375 | practice | 0.6406 | football | 5.17 | | |
| | | man jumping | 3.969 | lake | 0.2812 | leads the field | 0.625 | football team | 5.08 | | |
| | | pool | 3.344 | | | racecar driver leads | 0.2344 | friends playing | 3.844 | | |
| | | jump | 3.328 | | | driver leads | 0.1719 | young man throws | 3.188 | | |
| | | jumps | 3.047 | | | driver drives | 0.1719 | group of friends | 2.914 | | |
| | | water during sunset | 2.688 | | | drives | 0.09375 | pass | 2.64 | | |
| | | sunset | 2.469 | | | | | game against american | 2.219 | | |
| | | swimmers | 1.953 | | | | | man throws | 1.8125 | | |
| | | swimming | 1.75 | | | | | ball | 1.703 | | |
| | | woman is swimming | 1.234 | | | | | game | 1.625 | | |
| | | dog | 0.953 | | | | | throws a ball | 1.625 | | |
| | | water | 0.5312 | | | | | team | 1.3125 | | |
| | | man | 0.2344 | | | | | young man | 0.2344 | | |
| | | lake | 0.1562 | | | | | | | | |
| | | young man | 0.04688 | | | | | | | | |

Table 20: Results for B2T on the BAR dataset. Since B2T requires a validation set, we extracted ourselves a held-out validation split, stratifying with respect to the known class population distributions. Because of B2T's lack of a bias mining step, no misclassified examples could be found for the "Climbing" class, leading to a crash of the algorithm. Furthermore, there are no results for the class "Vaulting" because B2T filtered all the keywords out due to negative CLIP Scores. For the remaining classes, keep in mind that B2T's output consists of keywords that represent the opposite of the bias, which means that for the class "Fishing" the keywords "boy fishing", "beach", "fishing", "man fishing", and "lake" are wrong because they represent the actual bias or the class itself, and not the opposite of the bias as expected from B2T. The same can be said for the class "Racing" and the keywords "track", "race", etc. Overall, B2T's results on BAR appear to be wrong. For comparison, Tab. 26 shows SaMyNa's results on BAR.

**Differences in keyword filtering** An important novelty of SaMyNa is that it can find an embedding vector that represents the bias of the model in a certain class. This embedding vector is found by doing arithmetic operations between the embeddings of the captions, as explained in the paper. This means that we solve the problem of synonyms. In B2T there is a heavy filtering step before the ranking of the keywords that uses the YAKE keyword extraction algorithm. YAKE does not take into account the semantics of words, which means that it may filter out synonyms if they do not reach a certain frequency threshold individually. Conversely, our method does a very lightweight filtering of keywords before ranking, removing only very rare keywords that may result from captioning mistakes. After this lightweight filtering, we work entirely in the embedding space of the text embedder, which can account for all the synonyms and construct an embedding vector that represents the bias itself semantically. Keyword embeddings are then compared to the bias embeddings and ranked according to cosine similarity. Our heavy filtering is done after ranking by setting a similarity threshold. Evidence of B2T's filtering being too heavy is found in the full keyword rankings for the class "not blond" of CelebA in the supplementary material of B2T, where the keyword "woman" does not survive filtering and does not appear in the ranking. In particular, B2T selects the top 20 best keywords according to YAKE before ranking, while we discard keywords that do not appear in at least 15% of the captions for a given class and then aggregate the surviving keywords in a single pool of keywords that will be used for all classes.

**Symmetrical keyword rankings.** Additionally, our algorithm has the interesting mathematical property that for binary classification datasets, the ranking for one bias class is symmetrical with respect to the other class (because the two bias embedding vectors point in opposite directions, so the cosine similarity will give opposite scores).

**SaMyNa can be applied on images.** In Sec. A, we show the possibility of SaMyNa to work on other modalities without involving text. We use image embeddings instead of caption embeddings to produce the embedding vectors that represent the biases, and then we rank image patches instead of keywords according to the similarity of the patch to the bias embedding and we display this as a heatmap. This is a further novelty of SaMyNa with respect to B2T, showing that the underlying mechanism is fundamentally different (B2T only works with CLIP-like models and needs both images and text).

# E   Human Study Implementation Details

As for SaMyNa, we did not provide any guidance to the survey participants (in terms of bias features), so they did not have any prior knowledge about the kind of biases in each dataset. We report below the instructions provided to the participants in the survey:

> ***What represents a bias?*** *In the examples we will show, you will encounter different kinds of images. Some of them portray people, others portray animals, and others show activities performed by humans. In this questionnaire, we refer to every possible recurring feature as bias. Everything that shows a high correlation with the group should be then included in your answer (e.g. do not limit yourself to well-known biases such as gender).*

> ***How to fill out the form.*** *We will show you different groups of images. Each group has a common characteristic that we will highlight (hair color, type of bird, action name). For each group you must find up to 10 keywords, these keywords must refer to objects/features/characteristics or other elements of the images and must meet the following requirements:*
>
> - *The keyword must appear in MOST of the images of the group*
> - *The keyword must not be common to other groups in the same task (e.g. "person", "bird")*
> - *The keyword must not be the feature we highlighted about the group (if the group is about blond people, the keyword must not be "blond")*

# F    Automatic Procedure for Defining the $t_{\sf sim}$ Threshold

Even though we conceive SaMyNa as a supporting tool for which the final interpretation has to be performed by a human user, and that our filtering operations are intended to help tidying up our algorithm's output, we experimented with an automated way of finding per-class similarity thresholds $t_{\sf sim}(c)$ that are tailored to each specific model to be analyzed. The approach consists of obtaining a null distribution of cosine similarities by training the model under analysis on a modified version of the dataset where the labels have been randomized. In this way, we make sure that the original correlations between data and target labels are destroyed. The obtained model, while practically useless for the downstream classification task, will not be affected by particular biases as it has been trained on a random task. At this stage, we compute a null distribution of cosine similarities by extracting the learned class embedding and the keywords for each target class, computing an empirical distribution of the similarities between each keyword and the learned class embedding. The maximum cosine similarity emerging from this step will be chosen as the threshold for filtering keywords of a specific class.

We tested this alternative filtering approach on Waterbirds. A comparison between the obtained null distribution of cosine similarities, the distributions of the toy balanced version of Waterbirds (See sec. 5.2), and the original Waterbirds, can be found in Fig. 13.

As it can be seen, the balanced version of Waterbirds has a similar distribution to the randomized version of Waterbirds, which confirms that both were able to produce unbiased outcomes. The obvious difference is that for the randomized waterbirds dataset, as previously stated, the model produced is useless, but constructing the dataset is easier. The original Waterbirds' distribution, on the other hand, shows longer tails and 70% higher standard deviation. We exploit this fact to set the threshold for class $c$ to be equal to the maximum cosine similarity score of the keywords for class $c$ obtained from the model trained on the randomized dataset. These thresholds are then applied when generating the keyword rankings for the model trained on the original dataset. For the Waterbirds dataset, we obtained a threshold of 0.196 for the Landbird class, and a threshold of 0.320 for the Waterbird class. Comparing these thresholds with the scores shown in Tab. 22, you can see that the keywords that would be removed are "boat", "river", "midflight", "lake", and "flying" from the Waterbirds class, and no keyword would be removed from the Landbirds class.

Even though this method may seem better than setting a predefined threshold, it has two main disadvantages:

- It requires training the same model with the same procedure and hyperparameters on a randomized version of the dataset it was originally trained on, which is not always possible when using pre-trained models.

- It requires twice the compute time, since the entire pipeline must be run two times.

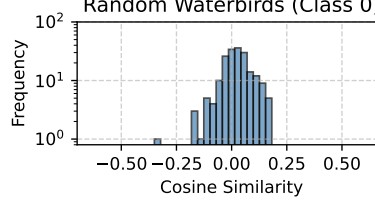 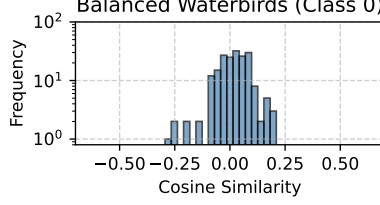 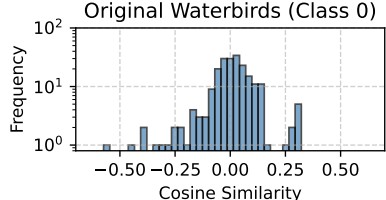

(a) Distribution of cosine similarities for the Landbirds class of the water-birds dataset with randomized targets.

(b) Distribution of cosine similarities for the Landbirds class of the balanced waterbirds dataset in Sec. 5.2.

(c) Distribution of cosine similarities for the Landbirds class of the original waterbirds dataset.

Figure 13: Comparison of the distributions of cosine similarities for the Landbird class of different versions of the waterbirds dataset. The histograms for the Waterbirds class are not reported because they are symmetrical, see Sec. D. As can be seen, the histograms from the random and balanced waterbirds version are similar in terms of variance (Std. Dev. 0.07 vs 0.075), while the original waterbirds dataset shows more variance (Std. Dev. 0.119), with longer distribution tails.

# G   How to Distinguish Harmful Biases from Target Concepts

A rare but still possible situation may be the one where target classes are highly stereotyped but not necessarily biased. As an ideal example, one might think of the case where one of the target classes is the *Zebra* animal: it is reasonable to expect SaMyNa to extract keywords such as `black-white`, `stripes`. In this particular case, it would not be correct to state that these concept keywords represent a harmful bias, but rather indicate that the model is relying on a very narrow set of concepts that are sufficient to correctly characterize the target class of interest. This occurrence may still be considered as a form of bias, but evidently not an unwanted one.

Even though this case may be rarely encountered, we show in this section how it is possible to leverage SaMyNa's pipeline to detect whether a model has captured a form of *benign* bias or an actually harmful one.

To simulate this corner case, we utilize the Colored MNIST dataset from Nam et al. (2020), where target classes are digits, and the bias is injected in the form of digit color, thus introducing a strong correlation between a specific color and a particular class. Consequently, a naive model ends up classifying according to color instead of shape.

In this custom experiment, we select the pool of correctly classified and misclassified samples for each class ($S^{correct}(c)$, $S^{misclass}(c)$) as follows: $S^{correct}(c)$ contains only samples belonging to class $c$ and sharing the same digit color (e.g., only red zeros for class *zero*). At the same time, $S^{misclass}(c)$ contains only samples from other classes having a different color (e.g., digits from 1 to 9, none of which are red). At this point, we may expect that all the keywords related to the correctly classified samples of a certain class ($S^{correct}(c)$) would be strongly correlated with the actual target and with the *red* concept, whereas samples from other classes present in $S^{misclass}(c)$ would not share some common feature causing them to be systematically mispredicted as, in this case, *zero*.

Hence, in practice, following the keyword extraction pipeline (Section 3.2.5), we obtain a set of potential bias keywords. At this point, whether a keyword is simply a target attribute and not a bias can be assessed by comparing two values: the average cosine similarity of the keyword's embedding with the caption embeddings of correctly classified samples, versus its average cosine similarity with the caption embeddings of misclassified samples.

In Table 21, we report the results of a comparison of such quantities when utilizing Colored MNIST as is (i.e., the biased case, "Original" in Table 21), or when we simulated the presence of a harmless bias ("Benign" in Table 21). The difference between the two aforementioned quantities is explicitly indicated in the two "$\Delta(\text{corr} - \text{misclass})$" columns, evidencing how in the *Benign* case this quantity is sensibly higher, indicating that the keywords correlate much more with correct predictions than with the model's mistakes.

| Class | Keyword(s) | Benign: | | | Original: | | |
|---|---|---|---|---|---|---|---|
| | | $\text{sim}(\text{emb}(kw), E^{\text{correct}}(c))$ | $\text{sim}(\text{emb}(kw), E^{\text{misclass}}(c))$ | $\Delta(\text{corr} - \text{misclass})$ | $\text{sim}(\text{emb}(kw), E^{\text{correct}}(c))$ | $\text{sim}(\text{emb}(kw), E^{\text{misclass}}(c))$ | $\Delta(\text{corr} - \text{misclass})$ |
| 0 | red | 0.4648 | 0.2589 | 0.2059 | 0.3978 | 0.3957 | 0.0021 |
| 1 | orange | 0.5088 | 0.1968 | 0.3120 | 0.4630 | 0.2975 | 0.1655 |
| 1 | red | 0.4648 | 0.2589 | 0.2059 | 0.3644 | 0.3351 | 0.0293 |
| 2 | yellow | 0.5795 | 0.2934 | 0.2861 | 0.5497 | 0.3736 | 0.1761 |
| 3 | green | 0.5200 | 0.2301 | 0.2899 | 0.4815 | 0.3912 | 0.0903 |
| 4 | green | 0.5235 | 0.2819 | 0.2416 | 0.4830 | 0.4403 | 0.0427 |
| 5 | blue | 0.4912 | 0.2687 | 0.2225 | 0.4796 | 0.4095 | 0.0701 |
| 6 | blue | 0.5104 | 0.2687 | 0.2423 | 0.4668 | 0.3824 | 0.0844 |
| 7 | purple | 0.5412 | 0.2482 | 0.2930 | 0.5106 | 0.3974 | 0.1132 |
| 8 | pink | 0.5016 | 0.1765 | 0.3251 | 0.4780 | 0.3812 | 0.0968 |
| 9 | pink | 0.5039 | 0.2307 | 0.2732 | 0.4601 | 0.3608 | 0.0993 |

Table 21: Average difference in cosine similarity between caption embeddings of correctly classified samples and an output keyword, compared to the average difference in cosine similarity with respect to misclassified samples of a specific class. This quantity can be an indicator of the presence of a benign bias rather than a harmful one.

# H   Outputs of SaMyNa

In this section, we provide the detailed output of our bias naming process for the experiments described in the main paper. We report the obtained keywords alongside their associated cosine similarity, in decreasing order of value, for Waterbirds (Tab. 22), CelebA (Tab. 23), BAR (Tab. 26), ImageNet-A (Tab. 24 and 25), on a completely balanced version of Waterbirds (Tab. 27), and for the ablation study on $k$ (Tab. 28).

| *Landbirds* | | *Waterbirds* | |
|---|---|---|---|
| Keyword | Cosine Similarity | Keyword | Cosine Similarity |
| forest | 0.335 | ocean | 0.540 |
| foliage | 0.334 | beach | 0.434 |
| tree | 0.327 | shore | 0.386 |
| stalks | 0.314 | water | 0.386 |
| branch | 0.312 | waves | 0.335 |
| trees | 0.299 | boat | 0.294 |
| branches | 0.296 | river | 0.268 |
| vegetation | 0.273 | midflight | 0.244 |
| | | lake | 0.244 |
| | | flying | 0.218 |

Table 22: Cosine similarity scores for keywords associated with the *Landbirds* and *Waterbirds* classes.

| *Not blonde* | | *Blonde* | |
|---|---|---|---|
| Keyword | Cosine Similarity | Keyword | Cosine Similarity |
| man | 0.45820 | woman | 0.39947 |
| male | 0.39175 | blonde | 0.34301 |
| | | mascara | 0.24959 |
| | | makeup | 0.24877 |
| | | lipstick | 0.22272 |
| | | eyeliner | 0.20065 |

Table 23: Similarity scores for the CelebA dataset.

| | *Crayfish* | | *Rhinoceros Beetle* |
|---|---|---|---|
| Keyword | Cosine Similarity | Keyword | Cosine Similarity |
| crab | 0.39954 | forest | 0.27621 |
| meal | 0.31551 | black | 0.24731 |
| crustacean | 0.28440 | branch | 0.20823 |
| food | 0.23936 | stands | 0.20675 |
| sandy | 0.20224 | | |

| | *Stick Insect* | | *Cockroach* |
|---|---|---|---|
| Keyword | Cosine Similarity | Keyword | Cosine Similarity |
| hand | 0.41610 | insect | 0.29116 |
| thumb | 0.39488 | creature | 0.27834 |
| finger | 0.36924 | darkcolored | 0.27402 |
| touch | 0.33881 | beetle | 0.26163 |
| plant | 0.31364 | dark | 0.24713 |
| plants | 0.30637 | flattened | 0.24523 |
| foliage | 0.28583 | grasshopper | 0.24468 |
| garden | 0.27045 | crustacean | 0.22505 |
| field | 0.22898 | darker | 0.22500 |
| leaves | 0.21886 | floor | 0.21054 |
| forest | 0.21871 | black | 0.20142 |
| holding | 0.21059 | | |
| grasshopper | 0.20454 | | |

Table 24: Similarity scores for the *crayfish*, *rhinoceros beetle*, *stick insect*, and *cockroach* classes from ImageNet-A.

| | *Nails* | | *Mushrooms* |
|---|---|---|---|
| Keyword | Cosine Similarity | Keyword | Cosine Similarity |
| metal | 0.37880 | plants | 0.25968 |
| frame | 0.27216 | foliage | 0.22674 |
| rusted | 0.25972 | orange | 0.20766 |
| wooden | 0.23891 | | |
| weathered | 0.23700 | | |
| wall | 0.22942 | | |
| snake | 0.20548 | | |
| black | 0.20246 | | |

Table 25: Similarity scores for the *nails* and *mushrooms* (fungi) classes from ImageNet-A.

| Climbing | | Diving | | *Fishing* | | *Racing* | | *Throwing* | | *Vaulting* | |
|---|---|---|---|---|---|---|---|---|---|---|---|
| Keyword | Cosine Similarity | Keyword | Cosine Similarity | Keyword | Cosine Similarity | Keyword | Cosine Similarity | Keyword | Cosine Similarity | Keyword | Cosine Similarity |
| climber | 0.55677 | scuba | 0.58300 | boat | 0.43818 | racing | 0.42294 | pitch | 0.54658 | vaulter | 0.57871 |
| cliff | 0.54837 | underwater | 0.56077 | fishing | 0.42546 | cars | 0.42024 | baseball | 0.48251 | vaulting | 0.50903 |
| climbing | 0.51882 | diver | 0.49010 | river | 0.42202 | car | 0.38508 | pitcher | 0.48111 | vault | 0.47589 |
| climb | 0.50305 | diving | 0.34910 | ocean | 0.39295 | track | 0.31835 | batter | 0.46329 | jump | 0.42575 |
| rock | 0.41596 | dive | 0.33087 | lake | 0.32940 | stadium | 0.28773 | mound | 0.44152 | midair | 0.37533 |
| steep | 0.36449 | ocean | 0.28606 | water | 0.31149 | race | 0.27802 | player | 0.42868 | pole | 0.36865 |
| ascent | 0.32149 | depths | 0.27261 | marine | 0.29890 | speeds | 0.26571 | throw | 0.37765 | athleticism | 0.32692 |
| rocky | 0.30365 | swimming | 0.26935 | waves | 0.25389 | speed | 0.26221 | athlete | 0.37310 | suspended | 0.28935 |
| ruggedness | 0.28319 | wetsuit | 0.26063 | underwater | 0.23570 | asphalt | 0.26199 | sport | 0.36638 | diving | 0.27763 |
| jagged | 0.26699 | marine | 0.25820 | wetsuit | 0.21951 | football | 0.25381 | playing | 0.36413 | bar | 0.26088 |
| ascending | 0.20509 | water | 0.22338 | scuba | 0.20644 | competition | 0.23812 | glove | 0.36378 | high | 0.24619 |
| | | | | | | pitch | 0.23165 | elbow | 0.34375 | fields | 0.23763 |
| | | | | | | baseball | 0.22954 | field | 0.33142 | arched | 0.23102 |
| | | | | | | batter | 0.22345 | football | 0.31316 | raised | 0.22868 |
| | | | | | | player | 0.21114 | game | 0.31113 | athlete | 0.22734 |
| | | | | | | team | 0.21087 | ball | 0.30598 | dive | 0.22547 |
| | | | | | | pitcher | 0.20611 | team | 0.29213 | flipper | 0.21940 |
| | | | | | | | | arm | 0.28914 | skill | 0.21579 |
| | | | | | | | | athleticism | 0.27865 | peak | 0.20534 |
| | | | | | | | | fields | 0.25888 | | |
| | | | | | | | | skill | 0.23882 | | |
| | | | | | | | | striking | 0.22815 | | |
| | | | | | | | | target | 0.22690 | | |
| | | | | | | | | focused | 0.22104 | | |
| | | | | | | | | main | 0.21313 | | |
| | | | | | | | | position | 0.21137 | | |
| | | | | | | | | action | 0.21031 | | |
| | | | | | | | | stadium | 0.21019 | | |
| | | | | | | | | arms | 0.20977 | | |
| | | | | | | | | positions | 0.20790 | | |
| | | | | | | | | pole | 0.20539 | | |
| | | | | | | | | activity | 0.20472 | | |
| | | | | | | | | casting | 0.20445 | | |
| | | | | | | | | stands | 0.20009 | | |

Table 26: Similarity scores for the BAR dataset.

| | Landbirds | | Waterbirds | |
|---|---|---|---|---|
| | Keyword | Cosine Similarity | Keyword | Cosine Similarity |
| | - | - | sea | 0.31901 |
| | | | midflight | 0.23266 |
| | | | seagull | 0.22417 |

Table 27: Ablation study on a completely balanced version of Waterbirds.

| k = 1 | | k = 5 | | k = 10 | | k = 25 | | k = 50 | |
|---|---|---|---|---|---|---|---|---|---|
| Landbirds | Waterbirds | Landbirds | Waterbirds | Landbirds | Waterbirds | Landbirds | Waterbirds | Landbirds | Waterbirds |
| midair: 0.18 ± 0.12
soars: 0.15 ± 0.11
tree: 0.07 ± 0.10 | water: 0.24 ± 0.00
seagull: 0.17 ± 0.12
fish: 0.17 ± 0.12
boathouse: 0.17 ± 0.12
seabirds: 0.15 ± 0.10
ships: 0.14 ± 0.19
vessel: 0.14 ± 0.19
ship: 0.13 ± 0.19
pier: 0.10 ± 0.15
lake: 0.09 ± 0.13
sea: 0.09 ± 0.13
crane: 0.07 ± 0.10
dock: 0.07 ± 0.10 | tree: 0.33 ± 0.02
forest: 0.33 ± 0.03
foliage: 0.31 ± 0.02
branches: 0.31 ± 0.02
trees: 0.30 ± 0.01
branch: 0.30 ± 0.02
stalks: 0.28 ± 0.02
bamboo: 0.26 ± 0.02
vegetation: 0.15 ± 0.11
leaves: 0.15 ± 0.11
deciduous: 0.09 ± 0.12
stalk: 0.08 ± 0.11 | water: 0.38 ± 0.02
beach: 0.37 ± 0.03
seagull: 0.31 ± 0.03
lake: 0.31 ± 0.02
waves: 0.27 ± 0.01
midflight: 0.27 ± 0.01
flying: 0.17 ± 0.12
ocean: 0.17 ± 0.24
sea: 0.16 ± 0.23
watermark: 0.16 ± 0.11
shore: 0.12 ± 0.18
river: 0.09 ± 0.13
seabird: 0.08 ± 0.12
horizon: 0.07 ± 0.10 | forest: 0.36 ± 0.02
foliage: 0.35 ± 0.01
branch: 0.34 ± 0.02
trees: 0.33 ± 0.02
branches: 0.32 ± 0.02
stalks: 0.32 ± 0.01
tree: 0.24 ± 0.17
vegetation: 0.19 ± 0.13
leaves: 0.16 ± 0.11 | ocean: 0.54 ± 0.02
beach: 0.44 ± 0.04
water: 0.37 ± 0.02
sea: 0.36 ± 0.26
waves: 0.33 ± 0.01
boat: 0.31 ± 0.01
shoreline: 0.27 ± 0.19
shore: 0.26 ± 0.18
lake: 0.26 ± 0.01
midflight: 0.24 ± 0.02
flying: 0.14 ± 0.10
coastal: 0.11 ± 0.16
river: 0.09 ± 0.13
foam: 0.09 ± 0.13 | tree: 0.36 ± 0.01
forest: 0.34 ± 0.01
foliage: 0.34 ± 0.01
trees: 0.33 ± 0.01
branch: 0.31 ± 0.02
branches: 0.30 ± 0.01
leaves: 0.24 ± 0.01 | ocean: 0.58 ± 0.01
beach: 0.51 ± 0.02
water: 0.39 ± 0.02
lake: 0.30 ± 0.01
waves: 0.23 ± 0.16
sea: 0.20 ± 0.28
seagull: 0.14 ± 0.10
horizon: 0.14 ± 0.10
rocky: 0.07 ± 0.09 | tree: 0.38 ± 0.02
forest: 0.37 ± 0.01
foliage: 0.35 ± 0.01
branch: 0.35 ± 0.02
leaves: 0.24 ± 0.01
branches: 0.21 ± 0.15 | ocean: 0.56 ± 0.01
beach: 0.52 ± 0.01
water: 0.37 ± 0.02
waves: 0.34 ± 0.02
midflight: 0.26 ± 0.00
flying: 0.22 ± 0.01
sandy: 0.15 ± 0.11
horizon: 0.07 ± 0.10 |

Table 28: Ablation study on Waterbirds, where we analyze the impact of the number of selected examples on the found keywords ($k$) and their respective similarity values.