# OpenReview forum: "Say My Name: a Model's Bias Discovery Framework"
_TMLR — Accepted by TMLR_

### Review · Reviewer_QtvF · 2025-08-13

**Summary Of Contributions:**

This paper focuses on the issue of biases which are the spurious correlations that the models over-rely on specific patterns that are non-representative of the real-world data distribution. To address this issue, the contributions of this paper are summarized as:

a.	This paper proposes a tool for naming biases in DL models that is text-based and usable both at training and at inference time.
b.	This paper validates the proposed approach on established benchmarks, finding biases well-known by the community.

**Audience:**

Yes

**Claims And Evidence:**

Yes

**Requested Changes:**

a.	In Section 1, this paper describes the issue of biases which are very interesting. I think that it is better to provide experiments to demonstrate that the models sometimes do over-rely on the biases, I, e., specific patterns that are non-representative of the real-world data distribution.

b.	In Section 1, Contributions 1, 2 and 3 could be merged as one.

c.	In Section 3.1, It is started that “Inspired by the Hinge loss function Rosasco et al. (2004), we can define a per-sample distance metric …”. However, I think it is quite different from the Hinge loss. Please describe it in details.

d.	I think Eq.1 is not rigorous. Considering $ \hat{y} _{t,n} = [0.7,0.2,0.1]$ and $y _{n}=2$, then Eq.1 is equal to 0.5, which indicates that the sample is misclassified into only label 1 with high probability. Considering another case, $ \hat{y} _{t,n} = [0.41,0.2,0.39]$ and $y _{n}=2$, then Eq.1 is equal to 0.21. But in this case, the sample is misclassified into labels 1 and 3 with high probability. Therefore, Eq.1 and the statement “the higher $d _{t,n}$ is, the most $M$ s confident in misclassifying …” are not rigorous.

e.	Is $D^{misclass} _{train}$ changing with the increase of iteration $t$? What is the range of $t$?

f.	In the first sentence of Section 3.2.1, why is there a $c$ in front of $D^{misclass}(c)$? Why do you use a cluster algorithm k-medoid? What are the differences between $D^{misclass}(c)$ ($D^{correct}(c)$) and $S^{misclass}(c)$ ($S^{correct}(c)$)? It is better to rewrite Section 3.2.1 for readability.

g.	In Section 3.2.4, how to achieve the embedding matrices?

h.	In Eq.3, it doesn’t need to be divided by 2.

**Strengths And Weaknesses:**

Strengths:

a. This paper is well-written and easy to follow.

b. I appreciate the extensive experiments.

c. I think the topic of biases is very interesting.

Weaknesses:

a.	My main concern is about the proposed method. I don’t capture the technique contribution of the proposed method. Meanwhile, the motivation of each step in the proposed method is confusing.

---

> ### Author Response · Authors · 2025-09-05
>
> We thank the reviewer for the time taken to review our work. Given the number of raised points, we will answer in two separate comments.
>
> a. We thank the reviewer for the suggestion. In Section A of the appendix, we show an explainability visualization utilizing a vision encoder. Here, leveraging the waterbirds dataset, we show how the model is focusing mostly on the background, which is not the semantic attribute of interest for the classification task and is indeed a bias attribute of this dataset. In the revised version of the manuscript, we will place part of this visualization in Section 1, to provide the reader with a direct example of the effect of spurious attributes on the classification behavior of a model.
>
> b. We thank the reviewer for the suggestion. We will merge the first three contributions into an individual one for improved clarity.
>
> c. We thank the reviewer for the comment.
>
> The distance-metric formalized in Equation 1 is indeed inspired by the Hinge Loss. However, we agree that more discussion on this matter is required to clarify that the inspiration comes from the objective of using a simple metric for characterizing how much a sample is confidently misclassified as another class, rather than the ground truth, as this may suggest either the absence of the bias attribute affecting its category, or the presence of a feature spuriously correlated with another target.
>
> The provided formulation could actually be simplified as
>
> $\max\left(0, \max\left(\widehat{y}\_{t,n}\right) - \hat{y}\_{t,n} \left(y_n\right)\right)$
>
> Where $\hat{y}\_{t,n}$ is the output of the model at the classification layer at epoch $t$ for the $n\text{-}th$ sample, while $\hat{y}\_{t,n}(y\_n)$ refers to the output neuron’s activation referring to the ground truth class of the $n\text{-}th$ sample.
>
> Here, we could consider the maximum confidence of the model as the “target” positive class, which we intend as the category for which the model is most confident in classifying a sample, whether the prediction is correct or not. The confidence for the ground-truth class can be thought of as the classifier output in the classical hinge loss formulation. In this way, we are computing a per-sample term that indicates “how far” the model is from turning its prediction towards the correct one (as when the sample is correctly classified, the above is just zero).
>
> The equation could be further simplified by dropping the outer max, as we work with one-hot-encoded ground truth vectors. We agree that the presented equation is quite different from the Hinge Loss and that its formulation could be simplified to fewer terms. However, we deemed the direct interpretability of the formula in terms of its practical aim to be more important, and as such, we expressed it as a piecewise function with clear condition cases. In the revised version of the manuscript, we will rewrite this paragraph to improve both motivation and clarity.
>
> d. We apologize for the lack of clarity. Here, the statement actually refers to how the model is confident in classifying an individual as a specific class (different from the ground truth). As such, in the second case provided in the example, we believe that the provided example shows precisely how Eq. 1 is intended to work. Specifically, here the sample would be classified as class 1, with the model also giving a comparable probability to class 3. As such, the model is still making an error, as in the first case, but it is an error made with “lower confidence” than the first example, where the model is again wrong in predicting class 1 for the sample, but it is also very confident in making this mistake.
>
> e. Yes, $D^{misclass}\_{train}$ can change during each iteration. However, the final $D^{misclass}\_{train}$ will depend solely on the model trained up to epoch $t^\*$ (see Section 3.1). Thus, the number of iterations $t$ can be set arbitrarily, given that we will consider only the model that during training maximizes the cumulated “misclassification distances” (Eq. 1 and 2). Once $t^\*$ has been determined, $D^{misclass}\_{train}$ no longer changes and is used in the rest of the pipeline.

---

> ### Author Response · Authors · 2025-09-05
>
> f. We thank the reviewer for highlighting some unclear points in the mentioned section. We will rewrite it to improve clarity and readability, while here we provide punctual responses to the reviewer’s doubts.
> - Why is there a c in front of $D\_{misclass}(c)$?
>     - We apologize for omitting a comma, which would have made this sentence less ambiguous. $D\_{misclass}(c)$ is simply the pool of samples misclassified as class $c$, and the $c$ in front is not part of the set name. A clearer version of the sentence could be “... and a pool of samples misclassified as class $c$, which we call $D\_{misclass}(c)$”.
> - Why do you use k-medoid?
>     - Considering that our aim is to extract pools of candidate samples to extract potential bias keywords, k-medoid is a natural choice for a clustering algorithm in this context. Given that correctly classified samples generally greatly outnumber the misclassified, we perform k-medoid rather than other algorithms exactly because it provides medoids (that can be referred to original samples) and not centroids (which may be arbitrary points in feature space). This provides us with a sample subset selection that can be fed to the subsequent phases of the algorithm. K-medoids is also useful because it provides better coverage in the sample selection, by using it we avoid selecting similar samples, which could happen with random selection.
>
> g. We thank the reviewer for this question.
>
> To obtain the embedding matrices $E^{correct}(c)$ and $E^{misclass}(c)$ we take each image sample from $S^{correct}(c)$ and $S^{misclass}(c)$ and we pass the images to a captioning model. The resulting captions are then fed into a text encoder that produces embeddings of $Z$ dimensions, we then stack together these embedding vectors (one for each sample) to create a matrix. We keep the separation between correctly classified examples and misclassified examples by storing the embeddings in the two, previously cited, matrices. We will add these details to Section 3.2.4 in the revised version of the manuscript.
>
> h. Yes, technically, the method may work without the division by 2, but we kept it for theoretical correctness since the equation represents an average of averages between all possible pairs of samples composed by 1 correctly classified sample, and 1 incorrectly classified sample. So when doing the inner average for the pair, we divide by 2.

---

### Review · Reviewer_aTpa · 2025-08-19

**Summary Of Contributions:**

- The authors propose a pipeline to name model biases. The pipeline has four major steps:
	1. **Pinpointing a particular model for analysis.** If one has access to a validation set, use the final trained model. If not, one can use the checkpoint of the model for which it has the greatest proportion of confidently misclassified samples.
	2. **Extracting samples.** For each target class, identify the samples which are correctly and incorrectly classified into that target class. Run k-medioids to subsample the correct samples, and select the top k most confidently misclassified samples.
	3. **Generating candidate bias keywords.** Caption each image with LLM. Extract keywords with custom pipeline.
	4. **Filtering bias keywords.** Generate per-class embeddings for each class by averaging per-image embeddings generated with a pre-trained text encoder. Get a set of keywords which occur often within each class, align with each class embedding up to a certain threshold.
- The authors then evaluate the performance of convolutional image classification architectures (ResNet-18/ResNet50) on four datasets previously studied in the debiasing literature: CelebA, BAR, Waterbirds, and ImageNet-A. All models are pre-trained on ImageNet-1K, and either fine-tuned (CelebA,BAR,Waterbirds) or tested directly (ImageNet-A) on these datasets of interest.
	- For WaterBirds, CelebA, and BAR, the authors demonstrate that their pipeline rediscovers known biases within the dataset, when evaluated only on the training dataset.
	- For ImageNet-A, the authors demonstrate that using their pipeline on a model at inference time extracts keywords which potentially represent biases.
	- Once again for WaterBirds, CelebA, and BAR, the authors combine their keyword extraction framework with GroupDRO. In particular, they obtain per-sample bias labels by using a CLIP zero-shot classifier to extract potential alignment between per-sample classification and per-class bias keywords.
-  Finaly, the authors present a variety of other analyses, showing the performance of SaMyNa on vision transformer models, using an unbiased dataset, a specific case where SaMyNa extracts non-bias keywords, correlation between keywords with human feedback on CelebA, BAR, and Waterbirds, as well as a set of other ablation studies.

**Audience:**

Yes

**Broader Impact Concerns:**

My concerns around the nature of SaMyNa's bias keywords relates to my concerns about the broader impact statement currently included. I will refrain from making additional recommendations regarding broader impact until the authors respond to my initial round of comments.

**Claims And Evidence:**

No

**Requested Changes:**

I have organized my requested changes to match the weaknesses described above.
## Keywords generated by SaMyNa
These changes are critical to securing my recommendation for acceptance.
- Please demonstrate if keywords generated by SaMyNa are truly reflective of model bias, and not dataset bias. As an example, if you were to use SaMyNa on Colored MNIST as in Nam et al. 2020, how would it handle models trained on the malignant vs. benign bias settings?
- It would be useful to understand how bias keywords are associated with correctly vs. incorrectly classified samples. For example, for how many correct vs. incorrect samples is "vaulting" is a keyword for $c=\text{vaulting}$ in the BAR Dataset?
- Is there some "null distribution" of cosine similarities that can be used to define keywords which are signficantly indicative of bias?
- Beyond these changes, please include a thorough discussion of the keywords generated by SaMyNa, and how they coincide with actual biases of the model. Additional discussion of how to identify real biases from SaMyNa keywords would be important as well.
## Training time bias naming paradigm
These changes are critical to securing my recommendation for acceptance.
- How are the keywords and cosine similarity scores generated from training time bias naming related to the keywords and cosine similarity scores generated from inference time bias naming, for Waterbirds and CelebA? How does this change as you consider different training epochs $t^*$?
- Please discuss how this bias naming paradigm relates to previous work around memorization and mislabeled examples (e.g. , such as Pleiss et al. 2020.)
- Please specify if in Table 1, comparisons are performed using Training time bias naming, or Inference time bias naming for SaMyNa + GDRO.
- The text states: `Notably, since both our method and B2T rely on GroupDRO for the final bias mitigation step, our advantage has to be imputed on a finer semantic bias discovery.` In Table 1, if I understand correctly the comparison between B2T+GDRO and SaMyNa + GDRO involves training on datasets of different sizes. Can you tell us how SaMyNa +GDRO performs on the same size training set as B2T+GDRO (i.e.g holding back the same held-out validation set from the training set)?
## General clarity
I have described my concerns with general clarity above in the weaknesses section. These changes would strengthen the work in my view.

# References
Pleiss, Geoff, et al. "Identifying mislabeled data using the area under the margin ranking." Advances in Neural Information Processing Systems 33 (2020): 17044-17056.

**Strengths And Weaknesses:**

# Strengths
- The authors suggest a novel setup to address the problem of bias mining without a validation dataset. To my awareness this idea is novel, and has the potential to be quite useful to detect biases in many real world datasets and settings where a validation dataset may not be readily available.
- I appreciate the authors' fairly thorough comparison of their work to B2T (Kim et al. 2024) in the main text and Appx. D, given the apparent similarities between the work on first reading. Ultimately, as both methods involve multi-step pipelines with differences at each step, it is difficult to do an exhaustive comparison, but I believe the discussion of the differences is useful.

# Weaknesses
I believe there are three main weaknesses of the paper: 1) ambiguity in nature of keywords generated by SaMyNa, 2) lack of detail for the training time bias naming paradigm, and 3) general clarity.

## Keywords generated by SaMyNa
The greatest weakness of this paper is in the description of what the keywords generated by SaMyNa actually represent. The stated definition of biases in this paper are *spurious correlations*, which the model uses as "shortcuts" for the network to infer a prediction rule on training data which does not generalize well to test data. Throughout the paper, there are prominent examples of keywords extracted by SaMyNa which do not represent biases in this sense, or for which the concept of bias has considerable ambiguity. These ambiguities reduce the utility and explainability of biases labeled by SaMyNa for end users.
- The most prominent examples are in Figure 4, Table 2, and Figure 8. Here, some of the most strongly scored keywords are the target classes themselves (e.g. `vaulting` for `vaulting`), categories which are semantic generalizations of the target classes (e.g. `insect` for `cockroach`), or apparently useful features of classification (`metal` for `nails`). This observation has two potential sources. In one case, **SaMyNa could be extracting keywords which represent only correctly classified examples**, leading to extraction of useful, non-spurious features as keywords. In another case, **components of the SaMyNa pipeline share the same biases as the model under consideration**, leading to incorrect assignment of "bias" keywords to the misclassified examples. There are two concrete issues that arise from this feature of SaMyNa.
  - 1. **Model vs. Dataset bias.** If SaMyNa extracts keywords which only appear in correctly classified examples, it is very difficult to distinguish between "benign bias" in the dataset, and biases which the model is actively using to classify the data. In the text, the authors state: `Unlike approaches that discover biases in the dataset, our focus is on naming the biases captured by the model under examination.` Consider the Colored MNIST dataset as studied in Nam et al. 2020. When training the model on the `color` feature, the `digit` feature is a benign bias (e.g. not used for classification). If the goal of SaMyNa is to capture model biases, as opposed to dataset biases, the `digit` feature should not be extracted as a keyword. It is not clear to me that this should be the case.
  - 2. **Ambiguity in bias classification.** In section 5.3, the authors assert that the extracted keywords do not resemble biases. Why is the `wall` keyword not considered a bias for the `Nails` class in Imagenet, while `rock` is considered a bias for the `Climbing` class in BAR? Both are contextual cues, and the choice to consider one as a bias but not the other appears arbitrary. If SaMyNa can extract truly useful features in addition to spurious correlations, there are many datasets where it is non-trivial for a human user to distinguish which is which, and it may be difficult to mine biases in the presence of many real features. In Table 2, is `glossy` a bias feature for `Cockroach`? Without the assurance that extracted keywords are at least as related to misclassified samples as correctly classified ones, it can be difficult to identify biases from keywords, even for expert users.

3. **Relative scale of cosine similarity.** I am also concerned with the interpretation of the cosine similarity score used to rank keywords. Across the different models and datasets studied, it is not clear how to interpret different values of the statistic in terms of the relative "strength" of biases present. For example, many biases in the `Landbirds` target class have a cosine similarity score $\sim 0.3$ in Figure 3. Simultaneously, in Figure 7, for the `Waterbirds` target class the keyword `sea` has a similarity score of 0.3, **even fit on the debiased dataset**, and is described as "low correlation" in the text. Without some kind of relative scale to calibrate the meaning of the cosine similarity score, it is not clear how to define what constitutes significant levels of bias for each application of SaMyNa.

My impression overall is that SaMyNa aggressively generates candidate bias keywords, but that it is not clear how to distinguish real biases from useful features of the data. While this does not appear to be an issue for worst group classification (e.g. Table 1), imprudent suggestion of real features as bias keywords has concerning implications for many of the real world applications that the authors mention in their work.

## Training time bias naming paradigm
There is relatively little discussion around the training time bias naming paradigm; in my opinion, one of the most interesting parts of the paper.
1. Despite the existence of prior work that discusses similar training-time bias mining (Nahon et al. 2023), it would be important to understand how bias mining from training in this manner is related to the more typical practice of bias identification from a validation set, and how sensitive this process is to the choice of iteration $t^*$.
2. It would also be useful to discuss how this work is related to other works which use similar metrics to identify mislabeled data, as opposed to spurious correlations.
3. It is unclear if the bias mitigation in section 4.3 is performed using keywords which are derived from the training set or the validation set for SaMyNa + GDRO.

## General clarity
- In the text, it is not clear if  $\text{k}$ is the mediod number, and $k$ the number of incorrect samples, or if $k$ is shared for both.
- the citation style of the paper is very distracting, without distinction between textual and paranthetical citations.
- In my personal opinion, creating an acronym out of an evocative title is not very informative, and does little to tell the reader how this methodology differs from other work that exists in the field.

---

> ### Author Response · Authors · 2025-09-12
> **Answer to Reviewer aTpa**
>
> We thank the reviewer for their thoughtful review and useful comments. Below, we provide detailed responses to their main concerns.
>
> **Keywords generated by SaMyNa**
>
> *Please demonstrate if keywords generated by SaMyNa are truly reflective of model bias, and not dataset bias. How would it handle models trained on the malignant vs. benign bias settings?*
>
> It is important to notice that keywords are extracted and ranked from captions generated from correctly and misclassified samples. As such, if a dataset has a bias that the model does not capture, its misclassifications are not caused by the presence of that specific spurious correlation. Thus, SaMyNa would not detect attributes (and hence, keywords) consistently shared between correct and misclassified samples from the same target class.
> To support our claim, in Table 2 of the main text, we reported keywords extracted from four classes of ImageNet, using two different transformer models. The resulting keywords are indeed different and with different similarities for the two models, regardless of the pretraining dataset being the same, and also differ from the ones presented in Figure 5. For instance, whereas the ResNet-50 presents strong correlations between insect classes and the keyword “hand/thumb”, the two transformer-based architectures analyzed in Section 5.1 (and Table 2) do not seem to be affected as well.
>
> Regarding the question on benign and malignant biases, in the case of benign biases, we can interpret them as semantic attributes that indeed describe the target category. As such, we can expect the model to rely on such attributes to actually correctly classify a sample in its target class, but not to provide a shortcut for samples belonging to other categories, and as such, would not be captured by SaMyNa.
>
> On the contrary, if that attribute ends up providing a shortcut for other target classes, SaMyNa would indeed identify a common pattern between per-class correct and incorrect predictions of the model, suggesting that the attribute is actually a malignant bias for the model.
>
> *Is there some "null distribution" of cosine similarities that can be used to define keywords indicative of bias?*
>
> We thank the reviewer for this question. Inspired by this comment, we designed a custom approach to indeed get a null distribution to be later used for computing a per-class similarity threshold indicative of bias. Specifically, as SaMyNa is unsupervised with respect to bias labels, we randomize target labels and replicate the entire pipeline of SaMyNa on a model trained on this randomized task.
> In this way, the cosine similarities corresponding to the extracted keyword can indeed be interpreted as a null distribution, considering that potential spurious correlations have been destroyed by the randomization of target labels. The maximum cosine similarity for each class is used as a threshold for filtering SaMyNa keywords on the original dataset. Hence, if a keyword's cosine similarity is higher than the computed threshold, it is considered indicative of a potential bias. We empirically validated the newly designed approach on waterbirds, obtaining the following list of keywords:
>
> class 0: forest (0.334), foliage (0.334), tree (0.327), stalks (0.313), branch (0.312), trees (0.299), branches (0.296), vegetation (0.273)
>
> class 1: ocean (0.539), beach (0.434), shore (0.386), water (0.385), waves (0.334)
>
> Using the following computed thresholds: Landbirds: 0.196, Waterbirds: 0.320
>
> Even if this method provides an automatic procedure for filtering the keywords, it comes with some drawbacks, as it requires an extra effort in terms of computation. Specifically:
> It requires training the same model with the same procedure and hyperparameters on a randomized version of the dataset it was originally trained on, which is not always possible when using pre-trained models.
> It requires twice the computation time, since the entire pipeline must be run twice.
>
> Finally, by inspecting the obtained similarities, we noticed how their histogram overlaps very well with the one obtained on the toy experiment reported in Section 5.2, where we manually constructed a small, unbiased version of the same dataset. The histogram for the original waterbirds dataset shows instead longer tails and 70% higher standard deviation. We added these histograms and the description of the new procedure for filtering keywords in the new Section F of the Appendix.
>
> *Include a thorough discussion of the keywords generated by SaMyNa, and how they coincide with actual biases of the model.*
>
> In our work, we designed a validation protocol based on SaMyNa’s generated keywords, comparing them with the ones provided by 20 independent human subjects, showing a systematic overlap (See Sections 5.4 and E).
> Regarding how to identify real biases, we can refer to the previous answer, thus extracting a threshold with the randomization of target classes, and then use it to filter bias keywords.

---

> ### Author Response · Authors · 2025-09-12
> **Answer to  Reviewer aTpa (#2)**
>
> **Training time bias naming paradigm**
>
> *More discussion on the training time paradigm, bias identification without a validation set*
>
> Whereas many papers in unsupervised model debiasing rely on a held-out validation set to perform bias identification (in some cases also assuming that a small held-out for bias-annotated data is at disposal), this is, in our opinion, a strong limitation in this context. The typical real-world case would not provide anything other than class annotations, and if the dataset at hand has been inadvertently acquired with a bias, we may not know if the validation set is biased as well (or biased in a different way). For instance, in BAR from Nam et al., no validation set is provided, and simply uniformly sampling a validation set is likely to result in an even more biased validation set, which, when used to regularize a model under mitigation, could actually push the learning in the wrong direction. Other cases (such as BFFHQ, which we do not analyze in this work) present a validation set composed of only bias-aligned samples. Working solely with the training set allows for more flexibility in this regard, dropping a strong assumption that is rarely met in reality.
> However, without some dedicated technique to avoid complete memorization of the training set, it is highly challenging to identify at which iteration a bias discovery model better separates bias-aligned samples from the few conflicting ones. In this regard, the choice of the iteration $t^* $ is indeed fundamental for any technique relying on a separate model for bias identification. Our contribution in this matter is how we automatically select a stopping iteration $t^* $ (Section 3.1). In our framework, the underlying intuition is to capture the moment during training where the model is trying to leverage the spurious shortcuts to perform its predictions without having yet memorized the individual samples. As we can expect that memorization will happen over time, we can store an intermediate model according to our defined metric that we can expect to produce informative sets of True Positives and False Negatives.
>
> *Please discuss how this bias naming paradigm relates to previous work around memorization and mislabeled examples (e.g. , such as Pleiss et al. 2020.)*
>
> We thank the reviewer for highlighting an interesting point of discussion. Memorization and noisy labels are indeed topics that are related to challenges in bias identification and mitigation. In a strongly biased dataset, from the perspective of a model under training, bias-conflicting samples can be almost considered as “mislabeled”. Consider a model that is fit to recognize a category where 95% of the training samples share the same characteristics, and 5% while representing the same category lack the same cue as the others (which at these ratios can be considered spurious). In this case, one could say that the model is right in not recognizing a conflicting sample in the same class as the aligned ones, because according to the training data, it is too different from the majority of the other samples. Different from the noisy labels setting, however, this is true only because the training set has been sampled with a bias and does not well represent the true data distribution. As also evidenced in the referenced paper, the training model will still be pushed to forcefully learn that a specific sample is from a target class, memorizing it in the long run.
>
> Techniques from the noisy label settings have already been successfully applied in the context of bias mitigation, with one of the most notable examples being the Generalized Cross-Entropy [1], which mitigates the learning signal from “harder” samples, which may correspond to actually mislabeled inputs.
>
> Our technique shares some intuition with the one employed in the referenced paper from Pleiss et al., e.g. the idea of taking a margin-like measure of how the model is confident in misclassifying a sample as another class to determine which samples are to be treated differently to improve generalization. However, there are key differences, such as the absence of a fake class, as we desire to find model shortcuts that produce systematic errors, rather than errors due to noise in the data.
>
> Nevertheless, we included this paper among our references.
>
> [1] Zhang, Zhilu, and Mert Sabuncu. "Generalized cross entropy loss for training deep neural networks with noisy labels." Advances in neural information processing systems 31 (2018).

---

> ### Author Response · Authors · 2025-09-12
> **Answer to Reviwer aTpa (#3)**
>
> **General Clarity**
>
> *How are the keywords and cosine similarity scores generated from training time bias naming related to the keywords and cosine similarity scores generated from inference time bias naming, for Waterbirds and CelebA? How does this change as you consider different training epochs?*
>
> We thank the reviewer for the question. Keywords and cosine similarities are generated following the same algorithm described in section 3.2 in both situations, the only difference is the input for the pipeline described in section 3.2, where the set of correctly classified and misclassified samples changes according to the model's performance at epoch t. So for the training time algorithm we use the set of correctly and misclassified samples found at the most biased epoch t*, while for the inference time algorithm we practically use a pre-trained checkpoint at whatever epoch it was saved, which is usually the last epoch or the best epoch according to performance metrics like accuracy (in our case, model weights available from the torchvision API). Regarding the outputs of an inference time analysis on Waterbirds and CelebA, we did not intentionally cover this use case, as these datasets are benchmarks typically used in model debiasing that we employed to evaluate our method in well-known but realistic scenarios. The inference time analyses we provided on ImageNet pre-trained models served as an additional example of how to use SaMyNa more “in the wild”. Nevertheless, we can expect that considering models trained for a different number of epochs (or different hyperparameters, regularization schemes, etc) may result in different outputs provided by SaMyNa. However, in the specific case of Waterbirds and CelebA, we could expect that a model trained up to convergence with plain ERM will be irrecoverably biased and may have completely memorized the training set. However, in the inference time analysis, we consider new and unseen samples (such as those coming from a validation or a test set), where the model would still make similar mispredictions due to the training data shortcuts, and make SaMyNa behave similarly.
>
> *The text states: Notably, since both our method and B2T rely on GroupDRO for the final bias mitigation step, our advantage has to be imputed on a finer semantic bias discovery. In Table 1, if I understand correctly the comparison between B2T+GDRO and SaMyNa + GDRO involves training on datasets of different sizes. Can you tell us how SaMyNa +GDRO performs on the same size training set as B2T+GDRO (i.e.g holding back the same held-out validation set from the training set)?*
>
> We thank the reviewer for highlighting a point that was unclear. The comparison in terms of bias mitigation is performed utilizing the same setting of B2T+GDRO, whereas the key differences between the two approaches depend on the process for obtaining bias keywords and subgroup pseudo-labels on the training set. We utilize fewer samples than B2T for Bias Naming (thanks to the bias mining and exemplars extraction described in Sections 3.1 and 3.2.1), and differently from B2T, we do not rely on a validation set. We would like to specify that both Waterbirds and CelebA have a predefined and separated validation set, which we do not use. The training set for the mitigated model coincides with the training set of B2T+GDRO.  However, for replicating B2T+GDRO on BAR, and only for the bias keyword extraction, we hold out ourselves a validation set, as it is not predefined in the dataset. This is because a separate validation set is required for B2T to extract potential bias-keywords. Then, bias identification as described in B2T is performed on the whole training set, and the mitigation process relies on the same training data as the other methods.
>
> *Please specify if in Table 1, comparisons are performed using Training time bias naming, or Inference time bias naming for SaMyNa + GDRO.*
>
> We apologize for the lack of clarity. In Table 1, all results refer to applying our method at training time + GDRO. We better specified it in the text and its related caption in the revised version of the manuscript.
>
> *In the text, it is not clear if $\text{k}$ is the mediod number, and $k$ the number of incorrect samples, or if is shared for both.*
>
> We apologize for the imprecise notation highlighting. $\text{k}$ in our case indicates the same cardinality in both cases - we refer to $\text{k}$ for the medoids (for the correctly classified samples) while  $k$ (italic) for the incorrectly classified. We have revised the paper carefully to ensure consistency in this regard, by always using the italic $k$.
>
> *The citation style of the paper is very distracting, without distinction between textual and paranthetical citations.*
>
> We thank the reviewer for this comment. We fixed the citation style throughout the paper. Now every citation should be surrounded by parentheses for improved readability

---

> > ### Comment · Reviewer_aTpa · 2025-09-20
> > **Thank you.**
> >
> > I thank the reviewers for their response to my comments. In my opinion, many of the revisions cited by the reviewers have substantially improved the quality of the work (especially around setting thresholds around bias scores and general clarity).
> >
> > However, my main concern around the keywords generated by SaMyNa still remains. The authors state in their response, regarding benign biases:
> > ```
> > in the case of benign biases, we can interpret them as semantic attributes that indeed describe the target category. As such, we can expect the model to rely on such attributes to actually correctly classify a sample in its target class, but not to provide a shortcut for samples belonging to other categories, and as such, would not be captured by SaMyNa.
> > ```
> >
> > There is ample evidence in the manuscript that this is not true. For example, in Table 5: SaMyNa consistently identifies synonyms for the target class (or the target class itself) as potential biases (see my original statement on `Keywords generated by SaMyNa` for more examples).
> >
> > We can speculate as to why this happens. We can imagine a case where there are no coherent biases for a given class: half of the samples assigned to a target class $c$ are correct (in $\mathcal{S}^{\text{correct}}(c)$), and the other half are randomly assigned to class $c$ from all the other target classes (in $\mathcal{S}^{\text{misclass}}(c)$). In this case, there may not be any coherent "biases", in the sense of spurious correlations represented in $\mathcal{S}^{\text{misclass}}(c)$). Then, however, there is no part of the existing pipeline that prevents features only represented in $\mathcal{S}^{\text{correct}}(c)$ from being extracted by SaMyNa. I suggested the experiment with Colored MNIST as a direct test of this hypothesis, and likewise I suggested a deeper analysis of how bias keywords are associated with correctly vs. incorrectly classified samples. Neither of these suggestions were adequately addressed, and so my principle concerns with this work remain unacknowledged.

---

> ### Author Response · Authors · 2025-09-23
> **Follow-up response**
>
> We thank the reviewer for the follow-up on this matter.
>
> From a methodological perspective, in the absence of coherent biases, as in the example described by the reviewer, and by examining SaMyNa's pipeline, we can expect captions for misclassified samples to be reasonably uncorrelated among each other, while those for correctly classified samples should mainly share the target attribute.
>
> In the specific case of Colored-MNIST, we can further expect that expressions like "handwritten digits" would be generally shared across all captions, regardless of whether they refer to correctly or misclassified samples.
>
> As such, when computing the learned class embeddings, we can expect common expressions to be filtered by Eq.4,  while caption embeddings for correctly classified samples would mainly share the target class. As captions from misclassified samples should be uncorrelated, we can further reasonably expect $E^*(c)$ (Eq.3 and Eq.4) to be farther from $E^{correct}(c)$ and $E^{misclass}(c)$ than in the case of the presence of bias, where the bias is shared across such embeddings.
>
> Thus, even if the target class may be in the extracted keywords, its cosine similarity (computed with respect to  $E^*(c)$) should not be comparable to the case of actual bias shared among correctly and misclassified samples. As such, we can expect the procedure we defined in the first response to the reviewer (i.e., the null distribution threshold) to successfully filter out these keywords.
>
> As an empirical validation of such an explanation, we performed the experiment requested by the reviewer on Colored-MNIST, taking samples from the unbiased test set (thus avoiding coherent biases, by construction).
>
> Here is SaMyNa’s output filtered using the threshold obtained from the null distribution:
>
>
> bias 0: N/A
>
> bias 1: N/A
>
> bias 2: N/A
>
> bias 3: N/A
>
> bias 4: 4 (0.39730)
>
> bias 5: N/A
>
> bias 6: 6 (0.31059)
>
> bias 7: N/A
>
> bias 8: N/A
>
> bias 9: N/A
>
> And here are the thresholds obtained from the null distribution:
>
> Class 0: 0.3267753810896348
>
> Class 1: 0.25450048559756966
>
> Class 2: 0.37327122463935236
>
> Class 3: 0.37929560721287103
>
> Class 4: 0.37581832254170133
>
> Class 5: 0.3094666707673007
>
> Class 6: 0.2578874251993024
>
> Class 7: 0.4758029976655722
>
> Class 8: 0.41164416611464394
>
> Class 9: 0.27297411872151567
>
> As we can notice, only in 2 cases out of 10 the target class keyword has a cosine similarity higher than the threshold. But still, the cosine similarity exceeds the threshold by a small amount (0.02 for class 4, and 0.05 for class 6). This may be just from random chance given by the small sample size used in this experiment.
>
> Furthermore, there are no output keywords other than those 2, which would suggest the absence of bias, which is correct, since we applied the algorithm on an unbiased case.
>
> Anyway, we would like to emphasize that the target class is trivial to filter, both automatically and manually, by the user of SaMyNa. An automatic filter could be embedding either the class name or a description of the class, and then subtracting this embedding from the learned class embedding for that class. This ensures that the target class and its synonyms are downranked. Doing this is allowed because the target class is known. We didn’t focus on target class filtering because the filtering is trivial, and yet our tool is supposed to be of assistance to a human user.
>
> We also emphasize that we adopt the expression “learned class” embeddings in our method, and not “bias” embedding, as they indeed represent the learned class concepts.  Then, if the learned class corresponds to the target class, we can conclude that there is no problem with the model, since it’s predicting the target class as expected, without biases.
>
> We hope that this response clarifies the reviewer's doubts on this matter, and we will add this experiment and considerations in the camera-ready of the manuscript, if accepted.
>
> We thank again the reviewer for all the comments and the useful suggestions.

---

> > ### Comment · Reviewer_aTpa · 2025-09-30
> > **Thank you.**
> >
> > Thank you for your additional follow up. This is not actually the experiment I suggested, as there is no benign bias to be discovered if you use the unbiased test set as opposed to the training set. While the target class is trivial to filter, there are many cases throughout the paper where synonyms or relevant features are not filtered (see my initial statement of concerns).
> >
> > I will summarize by saying that while the authors' response has significantly improved the paper (in particular, introduction of the cosine similarity as a baseline seems important), my major concerns around the actual meaning of the keywords extracted by SaMyNa still remain. I think that a significant revision of the paper is necessary to better align what is discussed simply as "bias" as the features which SaMyNa actually extracts, and in this way the claims of the paper are not supported by the evidence.

---

> ### Author Response · Authors · 2025-09-30
> **More clarifications and new experiment on benign bias**
>
> We apologize for misunderstanding the proposed experiment, and we thank the reviewer for engaging in additional discussion on our work.
> If we correctly understand, the main concern regards how to prevent (or at least identify) features from  $S^{correct}(c)$ from being detected by our method. In this response, we provide a reasonable procedure for identifying this specific case.
>
> First, we would like to provide additional clarification on how SaMyNa can be utilized to highlight cases where a bias to be considered “benign” is present. To this end, we interpreted the reviewer’s suggestions as follows: in Colored MNIST, for each class, we consider a pool of correctly classified samples $S^{correct}(c)$ which all share the same target class and the same bias attributes, while in $S^{misclass}(c)$ we have samples that do not share any coherent and systematic attribute with the target. For instance, all the samples in $S^{correct}(0)$ (class zero) would be red zeros, while in $S^{misclass}(c)$ we can find samples from all other classes presenting colors different from red. At this point, we may expect that all the keywords related to the correctly classified samples of a certain class would be strongly correlated with the actual target and with the “red” concept (as they share the same “bening” bias), whereas samples from other classes would not share some common feature causing them to be systematically mispredicted as, in this case, zero.
>
> Taking this into account, and connected to our previous response, to highlight that this particular bias falls into the “benign” definition, as pointed out by the reviewer, we can check whether the keyword is much more correlated with $S^{correct}$ than $S^{misclass}$.
>
> To achieve this, we measure the average cosine similarity between captions from correctly classified samples and the embedding of an output keyword. If we compare it with the average similarity between captions from misclassified samples and the same keyword, we can highlight when a keyword is ranked high due to the presence of a “benign” bias, as this would be reflected by a large difference between these two average similarities. The results of the experiment we conducted (see below) demonstrate that this is indeed the case, and that a simple heuristic based on the difference or ratio between these two quantities should be sufficient to cover the great majority of corner cases like the one described by the reviewer. In this experiment, we exploited our automatic threshold procedure based on null distribution to filter the bias keywords, prior to computing the cosine similarities. We will include this additional functionality in the final version of the manuscript, if accepted.
>
> Nonetheless, we would like to underline again that SaMyNa’s main purpose is to highlight, through human-interpretable keywords, which concepts a model is relying on for making its predictions, and a human end-user is supposed to evaluate if the found keywords represent a bias or not. Despite the automatic threshold procedure based on null distribution and the newly designed procedure for testing the potential “benign” nature of identified bias keywords, our main aim is to provide useful information for a human user, to support analysis of a model’s behavior, or for applying debiasing methods.
>
>
>
>
> | Class | Keyword(s) | Benign: $\text{sim}\left(\text{emb}(\text{kw}), E^{correct}(c) \right)$ | Benign: $\text{sim}\left(\text{emb}(\text{kw}), E^{misclass}(c) \right)$ | Benign: $\Delta(\text{corr} - \text{misclass})$ | Original: $\text{sim}\left(\text{emb}(\text{kw}), E^{correct}(c) \right)$ | Original: $\text{sim}\left(\text{emb}(\text{kw}), E^{misclass}(c) \right)$ | Original: $\Delta(\text{corr} - \text{misclass})$ |
> |:---:|:---|:---:|:---:|:---:|:---:|:---:|:---:|
> | 0 | red | 0.4648 | 0.2589 | 0.2059 | 0.3978 | 0.3957 | 0.0021 |
> | 1 | orange | 0.5088 | 0.1968 | 0.3120 | 0.4630 | 0.2975 | 0.1655 |
> | 1 | red | | | | 0.3644 | 0.3351 | 0.0293 |
> | 2 | yellow | 0.5795 | 0.2934 | 0.2861 | 0.5497 | 0.3736 | 0.1761 |
> | 3 | green | 0.5200 | 0.2301 | 0.2899 | 0.4815 | 0.3912 | 0.0903 |
> | 4 | green | 0.5235 | 0.2819 | 0.2416 | 0.4830 | 0.4403 | 0.0427 |
> | 5 | blue | 0.4912 | 0.2687 | 0.2225 | 0.4796 | 0.4095 | 0.0701 |
> | 6 | blue | 0.5104 | 0.2681 | 0.2423 | 0.4668 | 0.3824 | 0.0844 |
> | 7 | purple | 0.5412 | 0.2482 | 0.2930 | 0.5106 | 0.3974 | 0.1132 |
> | 8 | pink | 0.5016 | 0.1765 | 0.3251 | 0.4780 | 0.3812 | 0.0968 |
> | 9 | pink | 0.5039 | 0.2307 | 0.2732 | 0.4601 | 0.3608 | 0.0993 |
>
> Here, in some cases, the same color keyword results as output for more than one class. This is due to the captioner being unable to differentiate between colors of very similar shade (e.g. class 9 color digit is  fuchsia, class 3 chartreuse and class 5 is cyan)

---

### Review · Reviewer_LSBx · 2025-08-30

**Summary Of Contributions:**

This paper suggests using a Vision Language Model to textually describe representative and strongly misclassified image examples from a previously trained classifier in order to attach semantics to potential model biases. Representative keywords of class are created and then compared with embedded descriptions that have common features across all classes removed. The keyword embeddings and "unique feature" embeddings are then compared via cosine similarity. When similarity is high, it indicates that a keyword is strongly associated with that class. In cases where strongly associated keywords are not strictly relevant to the class, a bias exists.

The method does not require a validation set, is useable at training and inference time, is validated on several benchmarks and can be coupled with existing debiasing strategies.

**Audience:**

Yes

**Broader Impact Concerns:**

It is important to acknowledge that VLM's are not an objective source of truth when determining biases in models or data sets.

**Claims And Evidence:**

No

**Requested Changes:**

As per point 2 above, a quick search turns up recent work in this field:

https://arxiv.org/abs/2207.01056?utm_source=chatgpt.com
https://arxiv.org/abs/2305.15407?utm_source=chatgpt.com
https://arxiv.org/abs/2503.08368

It is important to point out that the observer could have its own biases, and make note of the literature on this, even if there is empirical evidence that the observer is objective enough to get results on benchmarks.

The authors should account for why--if they cannot include an analysis of the observer's biases--this paper provides enough evidence for their results before I can fully support the claims that are made.

**Strengths And Weaknesses:**

The method is straightforward and intuitive. This is a reasonable use of a VLM to attach semantic meaning to image components for analysis.

I see two weaknesses:

1) Determination of representative samples uses k-medoids for correct and a hinge-inspired metric for incorrect classifications (Eq. 1). This is reasonable. Due to reliance on decision layer outputs (which can assign high confidence to incorrect predictions), it might be better to use metrics such as Mahalanobis or even Euclidean distance at the embedding layer. Euclidean distance from class centroids may be simpler and equally if not more effective for representative samples as well, given the tendency toward Neural Collapse (NC) (Papyan et al. 2020) in classification networks. There are nuances here, such as class imbalances, etc. and I don't think the paper needs to be revised to include these, but I think it is important to conside NC collapse geometry when exploring distance metrics in vision models for classification tasks.

2) The VLM that describes images and embeds text is assumed to be objective, and I don't see any discussion on debiasing the observer.

---

> ### Author Response · Authors · 2025-09-05
>
> We thank the reviewer for the time taken to review our work. Here are our answers to the reviewer's points:
> 1) We thank the reviewer for the useful suggestions. Specifically, other works have already studied the same issue in higher-dimensional space, leveraging Euclidean distance [1]. Our method differentiates itself with respect to [1] as it directly works on the output classification layer, not high-dimensional with respect to features coming from inner layers. This leads to some computational advantages, having an impact as well on optimization and better t* time extraction. As an example, looking at performance on CelebA, we notice that our method achieves a Bias-Conflicting accuracy ~6% higher. We understand that the NC problem can be leveraged with an analysis in inner layers; however, as we look at output distribution, we are less sensitive to any impact it might have. We understand that, working in a lower-dimensional space, we lose information; however, our empirical findings back the hypothesis that bias-related information, if leveraged in the target task, can successfully be extracted even in such a low-dimensional space.
> 2) We thank the reviewer for the comment. As shown in [2], VLMs can indeed be biased, and such bias could influence the generated bias keyword. However, in our work, we validated the output of our pipeline by comparing it with bias keywords provided by human subjects. The experiment (see Section 5.4 and Section E in the appendix)  confirms the absence of preexisting biases coming from VLMs for the investigated datasets, showing a good overlap with SaMyNa’s provided keywords. Nonetheless, we agree with the reviewer that relying on pre-trained VLMs, possibly showing preexisting biases, is a limitation of the proposed method. In the revised version of the manuscript, we will add these considerations and state these limitations explicitly in a dedicated paragraph.
>
> [1] Nahon, Rémi, Van-Tam Nguyen, and Enzo Tartaglione. "Mining bias-target alignment from voronoi cells." Proceedings of the IEEE/CVF International Conference on Computer Vision. 2023.
>
> [2] D'Incà, M., Peruzzo, E., Mancini, M., Xu, D., Goel, V., Xu, X., ... & Sebe, N. (2024). Openbias: Open-set bias detection in text-to-image generative models. In Proceedings of the IEEE/CVF Conference on Computer Vision and Pattern Recognition (pp. 12225-12235).

---

### Author Response · Authors · 2025-09-12

We thank the reviewers for their thoughtful and insightful comments on our manuscript. Their feedback has been very helpful in helping us to clarify key aspects of our work and to strengthen the paper's overall contribution.

In the revised version of the manuscript, we have implemented several changes to address the points raised. Specifically, we have:
- **Improved Clarity and Discussion:** We have rewritten sections to clarify our methodology, including the rationale behind our distance metric, the use of k-medoids, and the generation of embedding matrices.
- **Addressed General Weaknesses:** To provide a clearer definition of the keywords generated by SaMyNa, we have explored a new procedure for establishing a null distribution of cosine similarities. This introduces the possibility of automatically defining a threshold for identifying keywords that are indicative of a potential model's bias. We have also added a dedicated paragraph acknowledging the limitations of potentially biased pre-trained models.
- **Corrected and Added Details:** We have clarified the experimental setup for our comparisons, confirming that all results in Table 1 use the training-time bias naming paradigm. We also added an explainability visualization to the introduction to better demonstrate how models over-rely on biases.

The updated manuscript includes blue track changes to better highlight our modifications.

---

### Decision · Action_Editor_rTAe · 2025-10-24

**Recommendation:** Accept as is

**Audience:**

Yes

**Audience Explanation:**

The problem of unsupervised bias discovery is of high interest to the TMLR audience. Many researchers would be interested in whether if one could use the proposed idea to discover new bias in the widely used models.

**Claims And Evidence:**

Yes

**Claims Explanation:**

This paper introduces "Say My Name" (SaMyNa), a novel, text-based framework for identifying semantic concepts learned by a deep learning model. Its primary contributions are its ability to operate without a pre-annotated validation set (usable during training or post-hoc) and to provide human-interpretable keywords for these learned concepts, which can then be used to identify model biases.

After a thorough discussion and significant revisions from the authors, a majority of reviewers (LSBx, QtvF) found the claims in the revised submission to be well-supported.

Reviewer aTpa, however, maintained a recommendation for rejection due to a remaining concern. To the best of my understanding, this is not an issue of correctness but an issue of framing. Reviewer aTpa was appropriately concerned about the interpretation of keywords extracted by the model, as the method could extract useful, non-spurious features ("benign bias") from correctly classified examples, not just spurious correlations. The authors responded to this by introducing additional procedures to help users identify keywords corresponding to such benign cases. I believe this disagreement stems from the interpretation of what SaMyNa's "keywords" represent, and I find that the authors' revisions and clarifications in the current version of the paper address this issue to a satisfactory degree.